# Increased levels of endogenous retroviruses trigger fibroinflammation and play a role in kidney disease development

Poonam Dhillon[1,2,4], Kelly Ann Mulholland[1,2,4], Hailong Hu [1,2], Jihwan Park [1,2], Xin Sheng [1,2], Amin Abedini [1,2], Hongbo Liu [1,2], Allison Vassalotti[1,2], Junnan Wu[1,2] & Katalin Susztak [1,2,3] ✉

Inflammation is a common feature of all forms of chronic kidney disease; however, the underlying mechanism remains poorly understood. Evolutionarily inherited endogenous retroviruses (ERVs) have the potential to trigger an immune reaction. Comprehensive RNA-sequencing of control and diseased kidneys from human and mouse disease models indicated higher expression of transposable elements (TEs) and ERVs in diseased kidneys. Loss of cytosine methylation causing epigenetic derepression likely contributes to an increase in ERV levels. Genetic deletion/pharmacological inhibition of DNA methyltransferase 1 (DNMT1) induces ERV expression. In cultured kidney tubule cells, ERVs elicit the activation of cytosolic nucleotide sensors such as RIG-I, MDA5, and STING. ERVs expressions in kidney tubules trigger RIG-I/STING, and cytokine expression, and correlate with the presence of immune cells. Genetic deletion of RIG-I or STING or treatment with reverse transcriptase inhibitor ameliorates kidney fibroinflammation. Our data indicate an important role of epigenetic derepression-induced ERV activation triggering renal fibroinflammation.

Chronic kidney disease (CKD) is the tenth leading cause of death worldwide[1]. According to the latest data by the Center for Disease Control and Prevention (CDC), more than 1 in 7, or 37 million people in the US, are estimated to have CKD[2]. CKD is the fourth fastest-growing cause of death, which is increasing in incidence worldwide. The mechanism of CKD is poorly understood limiting the development of new therapies.

Transposable elements (TEs) are highly repetitive strands of DNA that are mobilized by DNA intermediates (DNA transposons) or RNA intermediates (Retrotransposons) in the human germline and certain somatic tissues[3]. Retrotransposons can be further classified based on the presence of long terminal repeats (LTR) in the sequence. LINE (long interspersed nuclear elements) and SINE (short interspersed nuclear

elements) are non-LTR retrotransposons, and endogenous retroviruses (ERVs) are designated as LTR- retrotransposons[3]. With similar genomic organization to exogenous retroviruses such as the Human immunodeficiency virus (HIV) and Human T-lymphotropic virus (HTLV), the ERV genome contains *gag, pol,* and *env* genes that are flanked by 5′ and 3′ UTRs[4,5]. It is estimated that 8% of the human genome and 10% of the mouse genome are ERV elements[3,6]. Acquired mutations in ERV sequences during evolution made most viral elements inactive. Epigenetic repression by cytosine methylation and histone modifications are important for ERV silencing[7–11]. In human and mouse early embryos, ERV transcripts comprise up to 15% of the cell's transcriptome[12–14] signifying the importance of ERV expression during embryonic development[12,13,15–17].

[1]Renal, Electrolyte, and Hypertension Division, Department of Medicine, University of Pennsylvania, Perelman School of Medicine, Philadelphia, PA 19014, USA. [2]Institute for Diabetes, Obesity, and Metabolism, University of Pennsylvania, Perelman School of Medicine, Philadelphia, PA 19014, USA. [3]Department of Genetics, University of Pennsylvania, Perelman School of Medicine, Philadelphia, PA 19014, USA. [4]These authors contributed equally: Poonam Dhillon, Kelly Ann Mulholland. ✉e-mail: ksusztak@pennmedicine.upenn.edu

Recent studies indicate increased ERVs levels in a variety of human diseases. Increased ERV expressions are considered pathogenic in autoimmune diseases such as systemic lupus erythematosus[18,19], multiple sclerosis [20–23] and neurological disorders such as Alzheimer's, Parkinson's disease, schizophrenia, and bipolar disorders[24–26]. Human endogenous retrovirus K {HERV-K(HML-2)} is one of the most recently acquired ERV with ~3000 proviral remnants in the human genome[27–30]. A recent study showed that ectopic overexpression of HERV-K *env* in the brain contributes to motor neuron dysfunction in mice[31].

Reactivation of ERVs could elicit a strong immune response, which is best characterized in cancer cells and was discovered via the widespread use of cytosine methylation inhibitors[32–37]. Due to sequence and structural similarities of ERVs to exogenous viruses, ERVs can be recognized by cytosolic nucleic acid sensors (RIG-I, MDA5, TLRs, AIM2, IFI6, cGAS-STING, etc.) leading to strong antiviral innate immune response resulting in cytokine and chemokine release. An increase in ERV levels boosts the innate immune response in an immune-compromised tumor microenvironment[38,39] and likely plays a key role in the therapeutic success of cytosine methylation inhibitors.

The role of ERVs in fibrosis and kidney disease has not been explored. In this study, we mapped the genomic landscape of TEs and ERVs in healthy and diseased mouse and human kidney tissue samples. We found that expression of some ERVs were higher in fibrosis and correlated with kidney disease severity, kidney immune cell fractions (CD8 T cells, CD4 T cells, natural killer cells, proliferating lymphocytes, monocytes, dendritic cells, granulocytes, plasmacytoid dendritic cells, basophils, and macrophages), and nucleotide sensors. Loss of cytosine methylation and epigenetic derepression likely contributed to the increase in TE/ERV levels. We show that ERV expression in vitro activates the cytosolic nucleotide sensors, RIG-I and STING. Genetic deletion of RIG-I or STING or pharmacological targeting of the reverse transcriptase ameliorated kidney disease severity in mice.

## Results

### Expression of TEs and ERVs in diseased human and mouse kidney tissue samples

First, we quantified TE and ERV expressions in human kidneys. We analyzed RNA sequencing data from 240 microdissected human kidney tubule samples (Fig. 1a) including patients with or without hypertension, diabetes, and varying degree of kidney dysfunction. Comprehensive clinical and demographic information such as age, race, gender, diabetes, and hypertension status were collected for each sample (Supplementary Data S1). Using RepeatMasker to quantify repetitive elements, we detected 104,717 TEs in human kidney tubule samples including 31,103 LINEs, 50,069 SINEs, and 10,581 ERVs. All three ERV classes, ERVK ($n = 232$), ERV1 ($n = 2232$), and ERVL ($n = 7787$) were identified. ERVs showed the highest expression amongst the classes of TEs, followed by SINEs and LINEs (Fig. 1b). Amongst the ERV class, the ERVK family exhibited the highest total expression followed by ERVL and ERV1.

Next, we evaluated the relationship between the expression of TEs and kidney disease severity, such as fibrosis. Using a linear regression model adjusted for age, race, gender, diabetes, hypertension, batch, RIN, duplication rate, mitochondrial percentage, unmapped reads, and unique reads, we identified 2711 TEs whose expression showed a significant linear correlation with interstitial fibrosis (FDR <0.05) (Fig. 1c). The level of 1925 TEs positively correlated with fibrosis, while 786 showed a negative correlation. LINEs, SINEs, and ERVs were among the classes of TEs that positively correlated with fibrosis in human kidney samples (Fig. 1d and Supplementary Data S1). The relationship between specific TEs from the ERVK, ERV1, LINE, and SINE families with kidney fibrosis are shown in Fig. 1e.

Next, we quantified the expression of full-length ERV proviruses using HervQuant, which employs a database of 3174 autonomous ERV elements in 485 human kidney tubule samples (Fig. S1a and Supplementary Data S1). We found that recently integrated Class II (HERV-K) showed the highest expression followed by Class I (HERV-E and HERV-H) and Class III (HERV-L) (Fig. S1b). We identified 199 full-length ERVs whose expression correlated with interstitial fibrosis (FDR <0.05), 74 of which showed positive and 125 showed a negative association with fibrosis (Fig. S1c and Supplementary Data S3). Of the full-length ERVs that showed a positive association with fibrosis, 72.97% belonged to Class I, 20.27% to Class II, and 5.41% to Class III (Fig. S1d). Recently integrated HERV-H and HERV-K showed a correlation with fibrosis (Fig. S1e).

To explore TE expression profiles in mouse models of kidney fibrosis, we evaluated kidney RNA sequencing data from folic acid (FA) and unilateral ureter obstruction (UUO)-induced models of kidney disease (Fig. 1f). We identified 130 TEs with higher levels and 6 with lower levels in the FA model, and 560 TE with higher and 59 with lower levels in the UUO model (Fig. 1g). Seventy-five TEs had higher expression both in the UUO and FA models, while one had lower expression in both models compared to controls (Supplementary Data S4). Also, we found that Class I (ERV1) and Class II (ERVK) ERVs showed higher expression levels in diseased tissue compared to the control (Fig. S1f). We validated the transcript expression of the top ERVs fragments including *MuLV, RLTR4, MMTV, LTRIS, GypLTR3A, MuRRS4*, and *LTR45* in mouse kidney disease models using quantitative PCR (Fig. S1g).

In summary, here we report TE and ERV expression in human and mouse kidney samples and identify a large number of TEs and ERVs showing a correlation with disease severity.

### TE and ERV levels correlate with cytosolic nucleotide sensors

To identify cells that express TEs and ERVs in the kidney, we performed in situ hybridization in human kidney samples. We observed increased RNA levels of the HERV-K *gag-env* gene mostly in diseased tubule cells (Fig. 2b). Similarly, we also observed an increase in protein levels of HERV-K (polymerase) in CKD tubule samples (Fig. 2c, d).

We next wanted to understand whether increased ERV expression potentially contributes to disease pathology. The nucleic acid receptors can be divided into two main categories: immune sensing receptors, which include Toll-like receptor 3, TLR7, TLR8, TLR9, a retinoic acid-inducible gene I (RIG-I), melanoma differentiation-associated gene 5 (MDA5)[39], absent in melanoma 2 (AIM2), and cyclic GMP–AMP synthetase (cGAS); and nucleic acid receptors with direct antiviral activity, including double-stranded RNA (dsRNA)-activated protein kinase R (PKR), IFN-induced protein with tetratricopeptide repeats 1 (IFIT1), 2′−5′-oligoadenylate synthetase 1 (OAS1), ribonuclease L (RNase L), and adenosine deaminase acting on RNA 1 (ADAR1) (Fig. 2a). By analyzing the RNA-seq dataset, we observed a positive correlation with cytosolic nucleotide sensors and TE/ERV expression in human CKD samples, which was particularly strong between cGAS, STING, IRF3/7, and ISG20, and TE expression (Fig. 2e). We obtained similar results when full-length ERV expression was analyzed (Fig. S2a, b).

Further analysis of protein expression of RIG-I, MDA5, cGAS, STING, phosphoSTING (pSTING), TBK1, and phosphoTBK1 indicated an increase in RIG-I, pSTING, pTBK1 levels in kidneys obtained from patients with CKD (Fig. 2c, d). Immunostaining analysis indicated an increased tubule epithelial protein expression of RIG-I, STING, and pSTING in human CKD samples (Fig. 2c and S2c). Furthermore, transcript levels of *RIG-I, MDA5, CGAS*, and *STING* in our dataset of 240 microdissected human kidneys, strongly correlated with the degree of kidney fibrosis (Fig. S2d).

Similarly, we observed an increase in transcript levels of various nucleic acid sensors such as *Rigi, Mda5, Lgp2, Tlr4, Tlr7, Aim2, Cgas, Sting* (Fig. S3a), and activated RIG-I signaling pathway (Fig. S3b) in kidneys of two different murine CKD models. RNA sequencing analysis showed a positive correlation between TE expression and the expression of genes associated with cytosolic nucleic acid sensing (Fig. S3c).

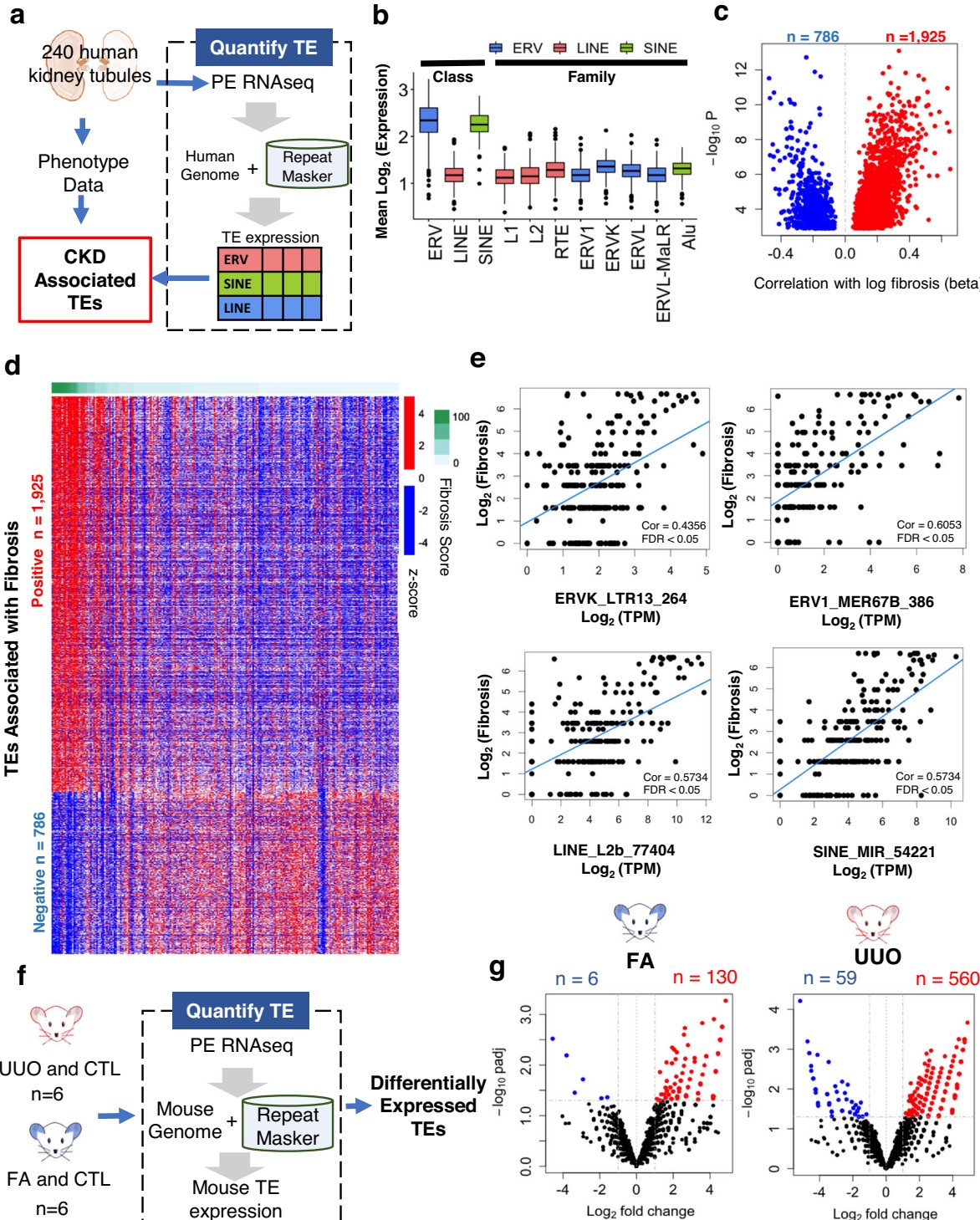

**Fig. 1 | Kidney expression of transposable elements (TEs) and their association with kidney disease severity in patients and mice. a** Our bioinformatic approach for TE discovery and quantification in 240 human kidney tubule samples. **b** Boxplot representing the relative (Mean log₂ TPM) expression of TE classes, endogenous retrovirus (ERV, blue), long interspersed nuclear elements (LINE, red), and short interspersed nuclear elements (SINE, green) and their families (ERV1, ERVK, ERVL, ERVL-MaLR, Alu, L1, L2, and RTE i.e., other retrotransposons). Center lines show the medians; box limits indicate the 25th and 75th percentiles; the bottom whisker indicates the fifth percentile and the top whiskers indicate the 95th percentile ($n = 240$ biologically independent samples). **c** Volcano plot of TEs showing association with kidney fibrosis in human kidney samples. The X-axis shows the beta values in the linear regression model (adjusted) for fibrosis. Y-axis shows the statistical significance (negative log $p$ value). Red dots show higher TEs and blue dots lower TEs

(FDR <0.05) in fibrosis. **d** Heatmap of TE ($z$-score) expression in human kidney samples. Each column is one kidney sample, and each row is one TE expression. Red indicates higher, blue indicates lower expression. The observed fibrosis in kidney samples is shown at the top. **e** The relationship between endogenous retrovirus K (ERVK), endogenous retrovirus 1 (ERV1), long interspersed nuclear elements (LINE) and short interspersed nuclear elements (SINE) levels (log₂ TPM counts, x-axis), and the degree of kidney fibrosis (y-axis, log₂ fibrosis). **f** Our quantification approach for TE expression in kidneys of murine renal disease models, FA (folic acid nephropathy), and UUO (unilateral ureteral obstruction). **g** Volcano plot of TEs showing association with kidney disease in mouse kidney samples. The X-axis shows the fold-change (control vs disease). Y-axis shows the statistical significance (negative log $p$ value). Red dots have higher TEs and blue dots have lower TEs (FDR <0.05) in fibrosis. Source data are provided in Supplementary Data 1–4.

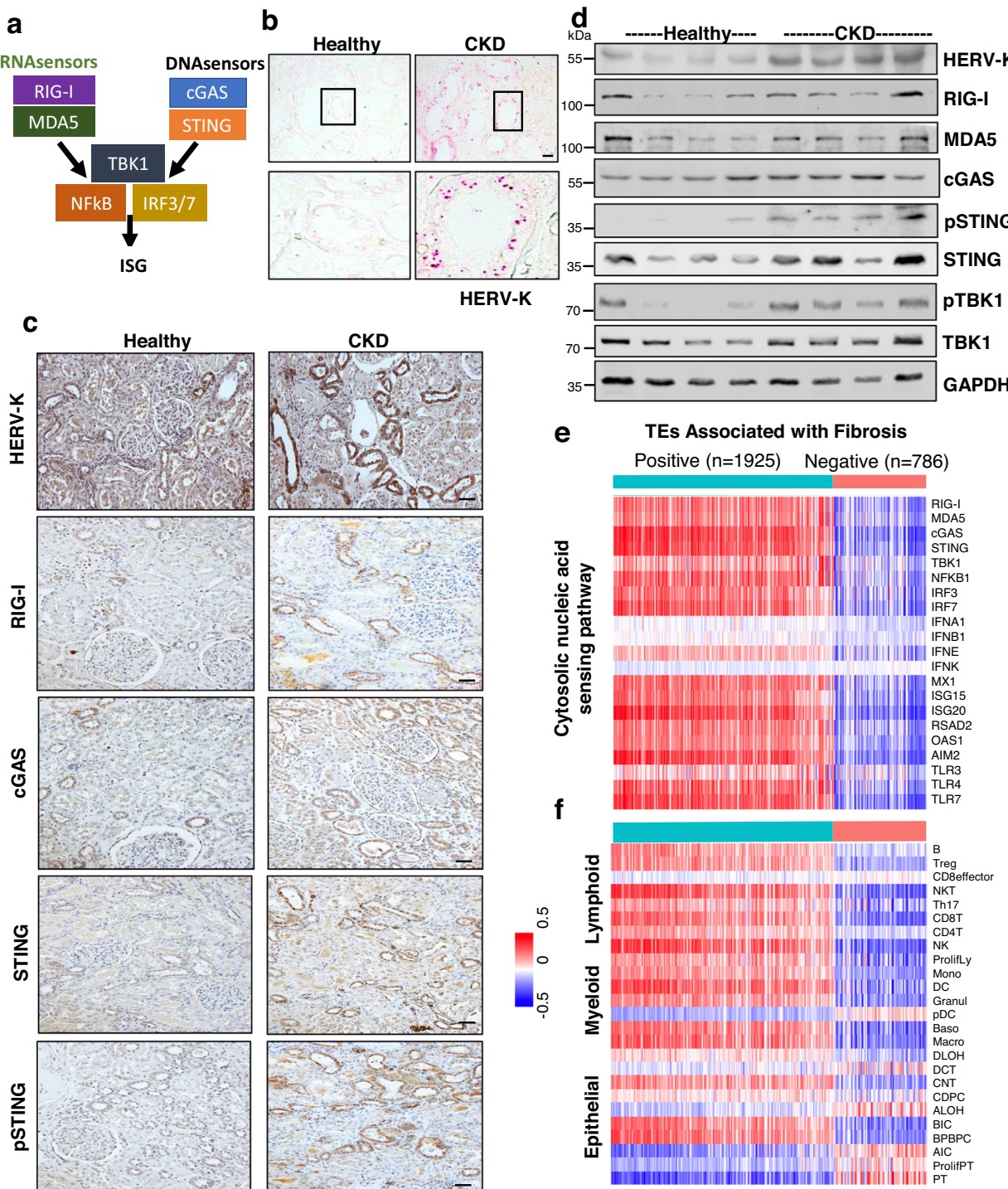

**Fig. 2 | Kidney HERV expressions correlate with nucleotide sensors and immune cell fractions. a** Schematic representation of cytosolic nucleic acid sensing pathway. **b** Representative images of in situ hybridization of HERV-K (pink) in healthy and chronic kidney disease (CKD) human kidneys. Scale bar, 20 μm. **c** Representative immunohistochemistry images of HERV-K, RIG-I, cGAS, STING, and pSTING protein expression in healthy and chronic kidney disease (CKD) human kidneys. Scale bar, 10 μm. **d** Western blot showing the expression of HERV-K, RIG-I, MDA5, cGAS, pSTING, STING, pTBK1, and TBK1 protein levels in healthy and human chronic kidney disease (CKD) samples. GAPDH was used as a loading control. **e** Heatmap of correlation coefficients (Pearson) transposable elements (TEs) and genes in the cytosolic nucleic acid sensing pathway in human kidney samples. Each column is one TE expression, and each row is one gene expression. Red indicates higher, blue indicates lower expression. **f** Heatmap of correlation coefficients (Pearson) of transposable elements (TEs) and estimated cell fractions in human kidney samples. Each column is one TE expression, and each row is one gene expression. Red indicates higher, blue indicates lower expression. CD8T CD8 T cells, CD4T CD4 T cells, NK natural killer cells, ProlifLy proliferating lymphocytes, Mono monocytes, DC dendritic cells, Granul granulocytes, pDC plasmacytoid dendritic cells, Baso basophils, Macro macrophages, DLOH descending loop of Henle, DCT distal convoluted tubule, CNT connecting tubule cells, CDPC collecting duct principal cells, ALOH loop of Henle, BIC type-B intercalated cells, BPBPC peripheral blood progenitor cells, AIC type-A intercalated cells, ProlifPT proliferating proximal tubules, PT proximal tubules. Data were representative of two independent experiments (**b**–**d**). Source data are provided as a Source Data file.

Higher protein levels of RIG-I, cGAS, pSTING, and STING were detected in injured tubules (Fig. S3d, e). We observed increased RIG-I expression in kidney tubule cells colocalized with high cytosolic dsRNA content in the UUO model (Fig. S3f), indicating the likely role of dsRNA activating the cytosolic nucleotide sensors.

Finally, we have also examined whether TE and ERV expression correlates with kidney immune cell fractions, such as likely eliciting an immune reaction. We estimated the fraction of 29 kidney cells via an in silico deconvolution method of bulk human kidney gene expression data using CIBERSORT[40] and cell-type markers obtained from single-cell RNA sequencing[41]. We observed a strong positive correlation between ERV levels and number of CD4, CD8, macrophages, and dendritic cells (Fig. 2f, Fig. S2e, and Supplementary Data S5). We validated the increase in immune cells (CD4, CD8, and macrophages) by IHC in healthy and diseased mouse (Fig. S4a) and human kidney sections (Fig. S4b). ERV level and tubule epithelial cell fractions (including proximal, distal tubules, and collecting duct), showed a negative correlation.

Collectively, we found that higher kidney tubule ERV levels correlated with an increase in cytosolic nucleotide sensor and immune cell fractions in human kidney samples.

## Ectopic expression of HERV-K in kidney tubule cells activates the cytosolic nucleotide sensing pathways

Our unbiased expression analysis and follow-up studies indicated an increase in the expression of HERV-K, which was detectable both at RNA and protein levels in kidney tubule cells. To understand the potential pathogenicity of HERV-K, the full-length provirus of HERV-K and control plasmid were transfected into primary tubule epithelial cells (PTECs) (for 24 h) (Fig. 3a and Fig. S5a). Ectopic expression of HERV-K lead to the increase in *Rigi, Mda5, Sting,* and *Aim2* transcripts while we did not detect a change in *Mavs, Tlr3, Tlr4, Tlr7,* and *Cgas* levels (Fig. 3b). Protein expression of RIG-I, cGAS, STING, pTBK1, and pIRF3 was elevated following HERV-K expression when compared to control vector. TBK1 and IRF3 activation was abrogated in RIG-I knockout (RIG KO) PTECs (Fig. 3c). Following HERV-K expression, we observed an increase in the expression of type I IFN response genes (*Ifnb, Ifne,* and *Ifnk*) and a much greater increase in RNA levels of Interferon-stimulated genes (*Mx1, Isg20, Rsad2, Oas1,* and *Bst2*) (Fig. 3d, e). The expression of type I IFN genes and ISGs were lower both in RIG KO and STING KO PTECs compared to wild-type (WT) transfected PTECs.

To further validate the pathogenicity of HERV RNA in PT cells, we used in vitro transcribed HERV RNA to transfect mouse primary tubule cells and human tubule (HKC-8) cell lines. We selected four HERV (HERV 3110, HERV 4321, HERV 4329, and HERV 915) that showed higher levels and one HERV (HERV 1132) that showed lower levels in CKD. Following the HERV RNA transfection we observed an increase in levels of cytosolic nucleotide sensors such as *Rigi, Mda5, Tlr7,* and *Aim2* followed by activation of downstream targets such as pTBK1 and pIRF3 (Fig. S5b, c). We also observed higher levels of *Ifnb,* and ISGs (*Isg15, Mx1, Rsad2, Bst2, Ifit1, Ch25h,* and *Isg20*) in tubule cells transfected with HERV RNA (Fig. S5d). Interestingly, we found HERV 4321 consistently showed robust antiviral response whereas HERV 1132 had minimal effect. This data highlighted the pathogenicity of kidney-specific HERV RNA that showed a strong positive correlation with fibrosis.

As the RNA-seq only provides a relative ERV abundance, we evaluated total dsRNA content in diseased kidneys by performing dsRNA-immunoprecipitation (IP). We found higher dsRNA content in the kidneys of UUO mice compared to sham (Fig. 4a). To show the pathogenicity of the dsRNA, we transfect PTECs with dsRNA isolated from UUO kidneys and found activation of RNA sensors (*Rigi, Sting,* and *Mda5*) (Fig. 4b) and increased expression of ISGs (*Mx1, Isg15, Oas1,* and *Bst2*) (Fig. 4c) and type I IFNs (*Ifnb, Ifne,* and *Ifnk*) which was abrogated in STING KO cells (Fig. 4d). Finally, to show the dsRNA-

mediated RIG-I activation we performed coimmunoprecipitation studies showing the dsRNA binding to RIG-I in diseased kidneys (Fig. 4e, f). To understand whether IFN might be responsible for ERV activation, we treated PTECs with mIFNβ. We found that Class I (*xMlv, eMlv, pMlv, mpMlv,* and *Mmtv*), Class II (*MervK, Musd, Iaps,* and *EtnII*), and Class III (*MervL* and *Malr*) ERVs and cytoplasmic nucleic acid sensors (*Rigi, Mda5, Mavs, Tlr7, Cgas, Sting,* and *Aim2*) levels were unchanged (Fig. S6a, b). Expression of ISGs (*Mx1, Isg15, Oas1,* and *Cxcl10*) were higher in kidney cells following mIFNβ treatment (Fig. S6c). Furthermore, we also tested the effect of lowering IFNG in the FA model of fibrosis by injecting mice with IFNG-neutralizing antibodies. IFNG lowering in mice had minimal effects on the expression of most of the ERVs and nucleic acid sensors (*Rigi, Mda5, Mavs, Tlr3, Tlr4, and Cgas*) except for *Tlr7* and *Sting* (Fig. S7a, b). Altogether, these results suggested that ERVs expression can activate RIG-STING pathways leading to IFN response in cultured kidney tubule epithelial cells.

## Epigenetic derepression is the likely cause of TE expression in kidney disease

DNA methylation and histone modifications have been shown to play a key role in repressing TE/ERV expression. We found HERV-K and HERV-H expression negatively correlated with DNMT1 and multiple other epigenetic regulators such as (*HDAC1, HDAC2, SETDB1,* and *TRIM28*) and TERT (Fig. S8a, b). Here, we investigated whether the change in DNA methylation could be responsible for the reactivation and expression of ERVs in human kidney tubules (Fig. 5a). We compared TEs expression and cytosine methylation levels using Illumina Infinium 850 K methylation arrays in 240 human kidney samples. We were able to obtain methylation information for 380 CpG probes that overlapped with the 2,711 TE sequences associated with fibrosis from our cohort (Fig. 5b and Supplementary Data S6). We identified 308 unique TEs with CpG probes. The majority of the overlapping probes (115 probes) negatively correlated with fibrosis levels, such as lower cytosine methylation was associated with higher TE expression (Fig. 5b, c), indicating that epigenetic derepression correlated with TE expression.

To further examine, whether loss of cytosine methylation is responsible for TE expression, we tested the effect of DNA Methyltransferase 1 (DNMT1) inhibition (DNMT1i) in PTECs. We observed an increase in several ERV families belonging to Class I ERVs (*eMlv, xMlv, mpMlv, pMlv,* and *Mmtv*), Class II ERVs (*MusD, Iaps, Etnii,* and *MervK*), and Class III ERVs (*MervL* and *MaLR*) in PTECs treated with the DNMT1i; 2-deoxy-5-azacytidine (Fig. 5d and Fig. S9a).

Treatment of cells with AZA also lead to the activation of the antiviral pathways such as *Rigi, Mda5, Tlr7,* and *Sting* and to a lesser degree of *Cgas* and *Aim2* (Fig. 5e). Protein levels of RIG-I, cGAS, STING, pTBK1, and pIRF3 were higher in AZA-treated cells and increased dsRNA levels showed co-localization with RIG-I protein, indicating the activation of the pathway (Fig. 5f and Fig. S9b). Furthermore, DNMT1i treatment was associated with an increase in the expression of IFN genes (*Ifnb, Ifne,* and *Ifnk*) (Fig. S9c) and ISGs (*Mx1, Isg20, Ch25h,* and *Rsad2*) (Fig. 5g). The activation of TBK1, IRF3, IFN, and ISGs were dependent on STING and RIG-I, as deletion of either of the sensors ameliorated the increase in expression of ISGs and IFNs (Fig. 5f, g). Our results indicate a negative correlation between cytosine methylation and ERV expression in human kidneys, furthermore DNMT1i treatment of cultured tubule cells induced the activation of IFN and ISG via STING and RIG-I.

## Loss of methylation induces ERVs expression and contributes to renal fibroinflammation

To further delineate the role of DNMTi-induced ERV expression in kidney disease development, we generated mice with tubule-specific conditional inducible DNMT1 deletion, using the tubule epithelial cell-specific doxycycline-inducible (Pax8rtTA) system. Pax8rtTA (tetracycline-responsive promoter element) transgenic mice were crossed

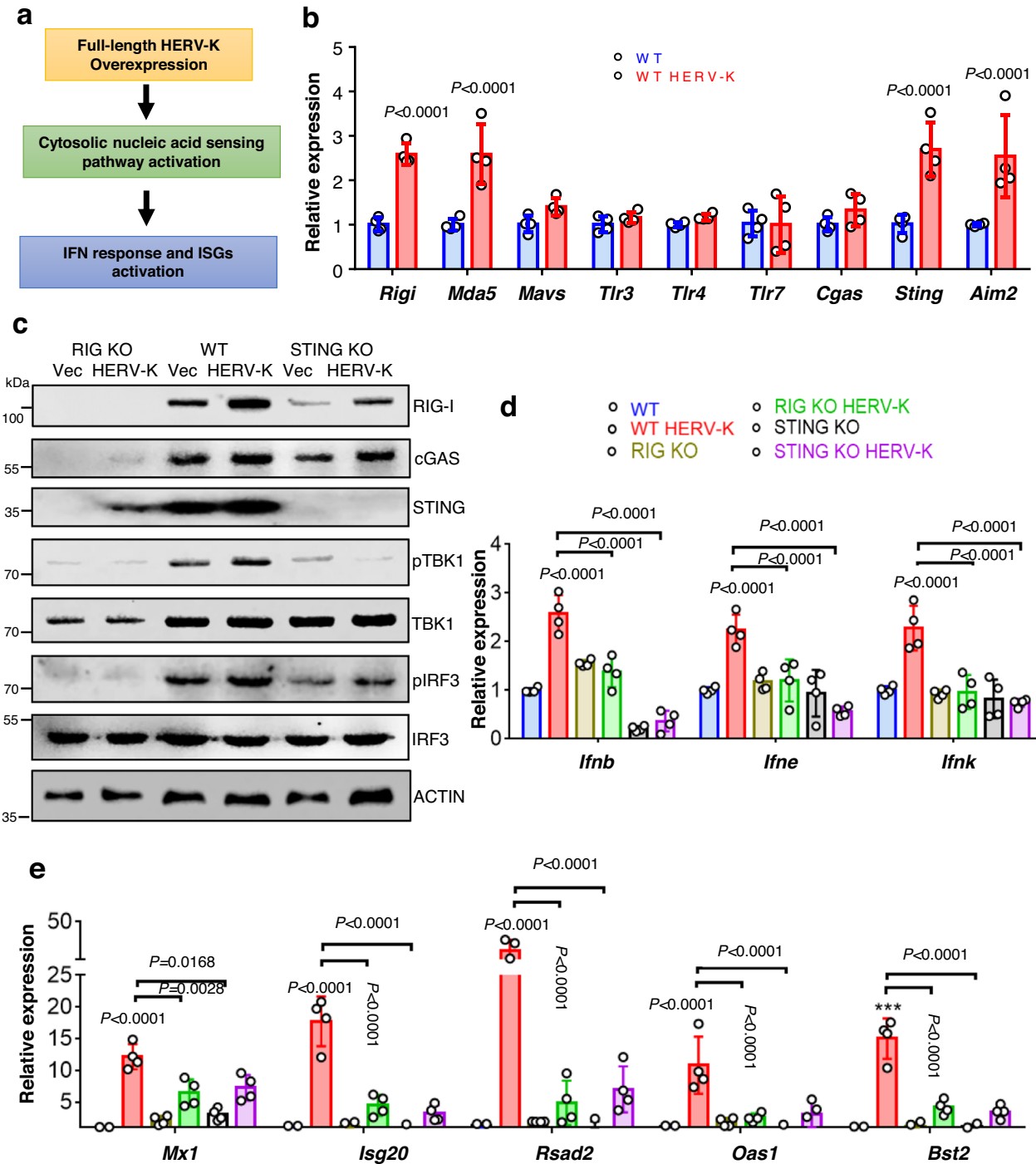

**Fig. 3 | HERV-K expression activates nucleotide sensing pathways in proximal tubule cells. a** Experimental hypothesis. **b** Relative mRNA levels of cytosolic RNA sensors (*Rigi, Mda5, Mavs, Tlr3, Tlr4,* and *Tlr7*) and DNA sensors *Cgas, Sting,* and *Aim2* in WT PTECs transfected with HERV-K (red) or Vector control (blue) (*n* = 3 in each). **c** Western blot showing representative images of RIG-I, cGAS, STING, pTBK1, TBK1, pIRF3, and IRF3 protein levels in WT, RIG KO, and STING KO PTECs transfected with HERV-K/Vector. ACTIN was used as a loading control. **d** *Ifnb, Ifne,* and *Ifnk* RNA expression in WT (Vector: blue, HERV-K: red), RIG KO (Vector: yellow, HERV-K: green), and STING KO (Vector: black, HERV-K: magenta) PTECs transfected with HERV-K or Vector control (*n* = 3 in each). **e** Relative mRNA levels of Interferon-stimulated genes (*Mx1, Isg20, Rsad2, Oas1,* and *Bst2*) in WT, RIG KO, and STING KO PTECs transfected with HERV-K/Vector (*n* = 3 in each). Data were presented as mean ± s.e.m. and all data were analyzed using a one-way ANOVA followed by Tukey post hoc test for multigroup comparison (**b**, **d**, **e**). Data were representative of two independent experiments (**c**). Source data are provided as a Source Data file.

with TRE/Cre (tet responsive Cre) and DNMT1$^{F/F}$ mice to generate Pax8$^{rtTA}$-TRE/$^{Cre}$-DNMT1$^{F/F}$ mice. Expression of DNMT1 was lower in kidneys of Pax8$^{rtTA}$-TRE/$^{Cre}$-DNMT1$^{F/F}$ mice compared to WT mice 7 days following doxycycline administration (Figs. S10a, S11a). Pax8$^{rtTA}$-TRE/$^{Cre}$-DNMT1$^{F/F}$ mice did not show renal abnormalities at baseline (however we did not analyze mice after prolonged aging). As DNMT1 is a de

novo methyltransferase, we reasoned that the lack of phenotype is likely related to the low cellular turnover of kidney tubule cells. We therefore induced injury by FA administration or by UUO surgery in DNMT1$^{F/F}$ and Pax8$^{rtTA}$-TRE/$^{Cre}$-DNMT1$^{F/F}$ mice (Fig. 6a and Fig. S11). Increased ERVs RNA levels: Class I ERVs (*eMlv* and *xMlv*) and Class II ERVs (*EtnII*) were detected in Pax8$^{rtTA}$-TRE/$^{Cre}$-DNMT1$^{F/F}$ mice at baseline

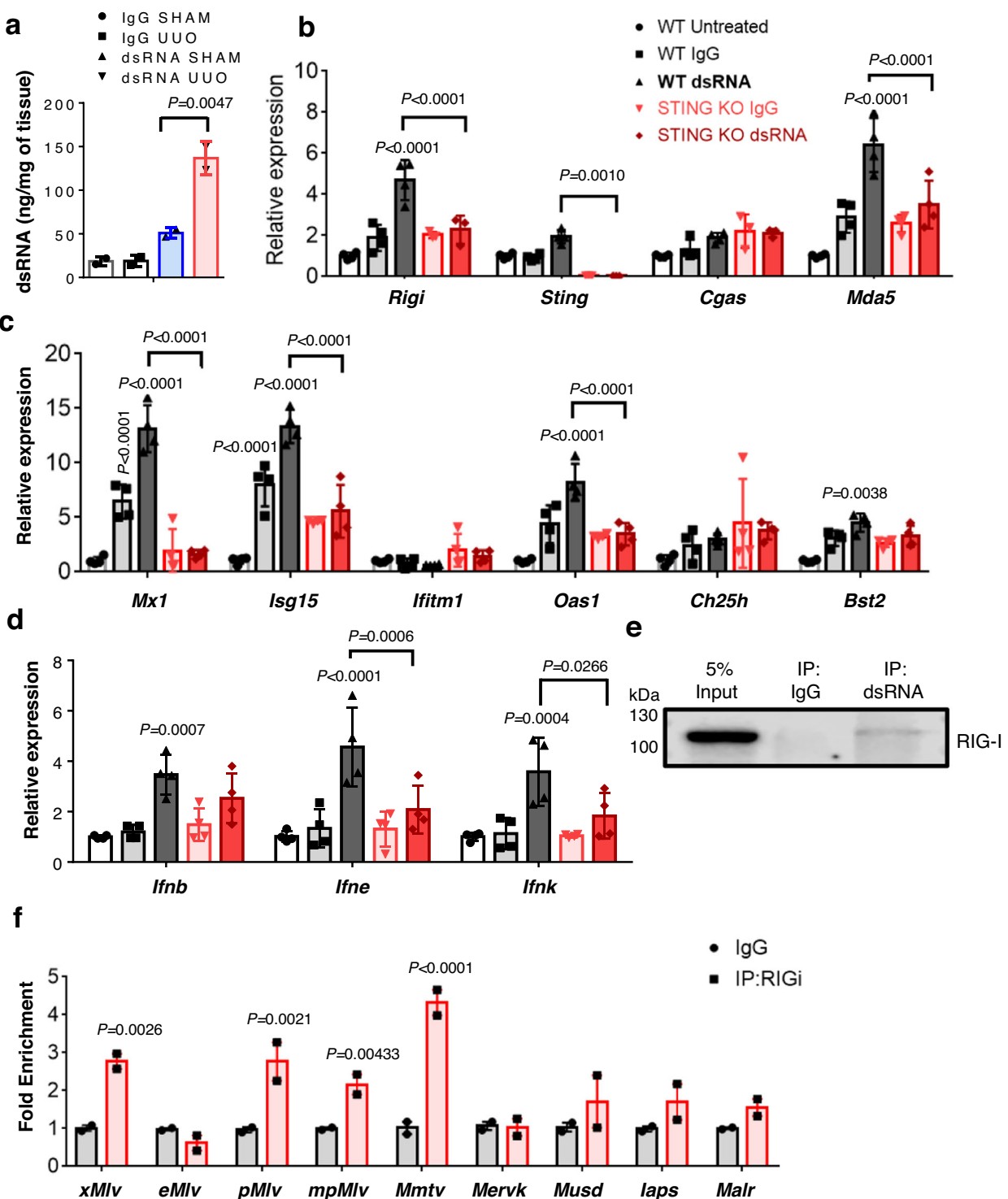

**Fig. 4 | Increased dsRNA content in UUO mice kidneys elicits innate immune response via RIG-I. a** dsRNA content in SHAM and UUO mice kidneys. Data were representative of two independent experiments (IgG, Black; dsRNA SHAM, blue; dsRNA UUO, red). **b** Relative mRNA levels of cytoplasmic nucleic acid sensors (*Rigi, Sting, Cgas*, and *Mda5*) in WT and STING KO PTECs untreated or transfected with immunoprecipitated RNA using IgG or dsRNA antibody ($n = 4$ in each). **c** Relative mRNA levels of ISGs (*Mx1, Isg15, Ifitm1, Oas1, Ch25h*, and *Bst2*) in WT and STING KO PTECs transfected with immunoprecipitated RNA using IgG or dsRNA antibody ($n = 4$ in each). **d** *Ifnb, Ifne*, and *Ifnk* RNA levels in WT (untreated, white; IgG, gray; dsRNA, black) and STING KO (IgG, pink; dsRNA, red) PTECs transfected with immunoprecipitated RNA using IgG or dsRNA antibody ($n = 4$ in each). **e** CO-IP

showing the interaction between dsRNA and RIG-I. Kidney lysates of mice with UUO injury were used for immunoprecipitation using IgG control and dsRNA antibody followed by western blot using RIG-I antibody. **f** Reverse CO-IP validating the interaction between RIG-I and ERVs. Kidney lysates from mice with UUO injury were used for immunoprecipitation using IgG control (gray) and RIG-I antibody (pink) followed by RNA isolation and qRT-PCR. The graph shows enrichment in Class I ERVs (*xMlv, pMlv, mpMlv*, and *Mmtv*), Class II (*Musd* and *Iaps*), and Class III (*Malr*) bound with RIG-I protein. Data were representative of two independent experiments. The data were represented as mean ± s.e.m. and all data were analyzed using a one-way ANOVA followed by Tukey post hoc test for multigroup comparison (**b**–**e**). Source data are provided as a Source Data file.

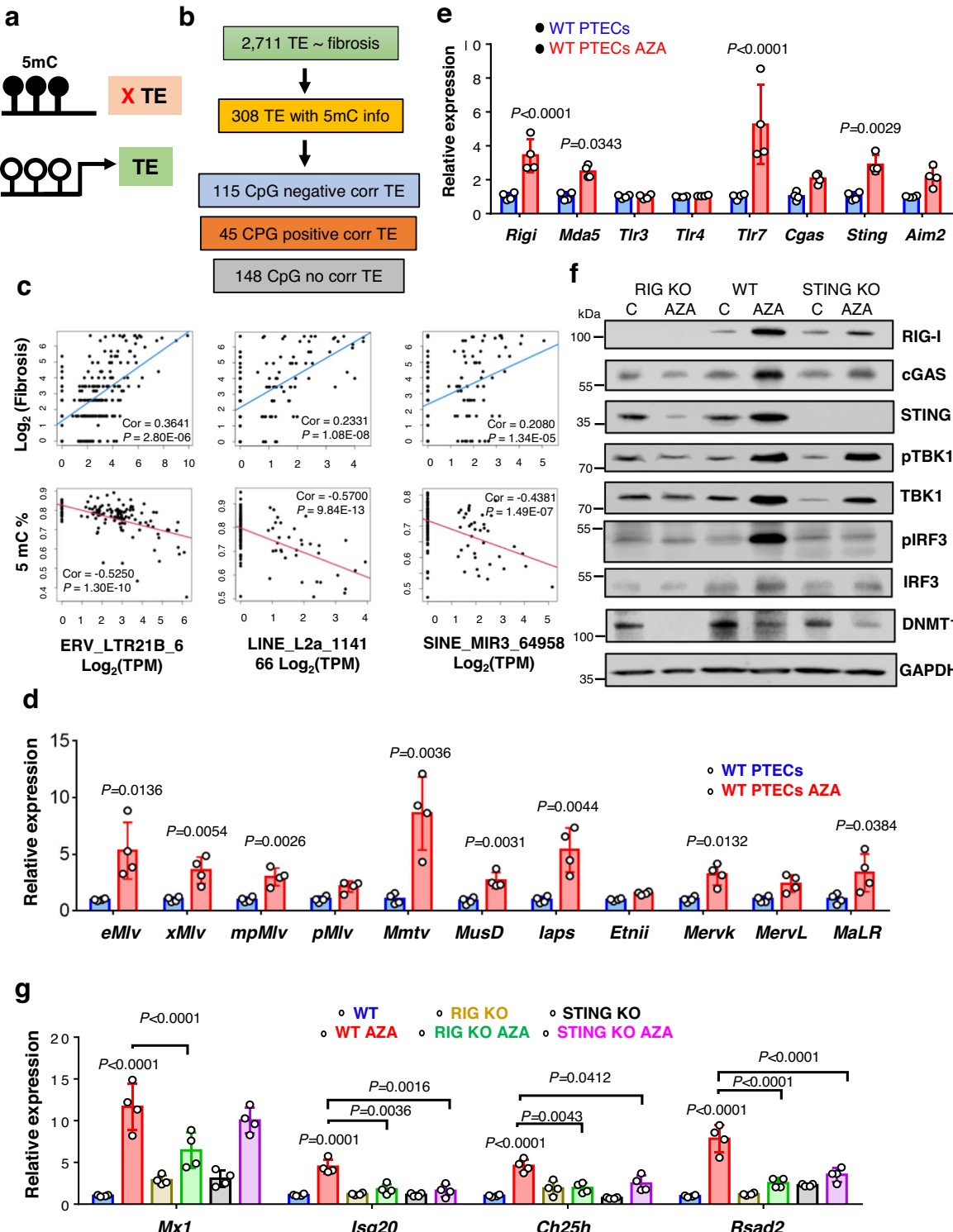

**Fig. 5 | The relationship between cytosine methylation and TE/ERV expression in human kidney samples. a** Experimental hypothesis. **b** Number of analyzed CpGs overlapping differentially expressed TEs. **c** The relationship between endogenous retrovirus K (ERVK), endogenous retrovirus 1 (ERV1), long interspersed nuclear elements (LINE) and short interspersed nuclear elements (SINE) levels (log₂ TPM counts, x-axis), the degree of kidney fibrosis (log₂ fibrosis, y-axis), or cytosine methylation level (5mC beta, y-axis). **d** Relative expression levels of Class I ERVs (*eMlv, xMlv, mpMlv, pMlv,* and *Mmtv*), Class II ERVs (*MusD, Iaps, Etnii,* and *Mervk*), and Class III ERVs (*MervL* and *MaLR*) in WT PTECs treated with AZA (5-azacytidine 0.25 μM) (red) or DMSO (blue) (*n* = 4 in each). **e** Relative expression of cytosolic RNA sensors (*Rigi, MdaS, Tlr3, Tlr4,* and *Tlr7*) and DNA sensors (*Cgas, Sting,* and *Aim2*) in WT PTECs treated with AZA or DMSO (*n* = 4 in each). Data were presented

as the mean ± s.e.m and analyzed by two-tailed unpaired Student's *t*-test (**d**, **e**). **f** Western blot RIG-I, cGAS, STING, pTBK1, TBK1, pIRF3, IRF3, and DNMT1 protein levels in WT, RIG KO, and STING KO PTECs treated with AZA or DMSO denoted as Control "**C**". GAPDH was used as a loading control. Data were representative of three independent experiments. **g** Relative mRNA levels of ISGs (*Mx1, Isg20, Ch25h,* and *Rsad2*) in WT (DMSO, blue; AZA, red), RIG KO (DMSO, yellow; AZA, green) and STING KO (DMSO, black; AZA, magenta) PTECs treated with AZA/DMSO (*n* = 4 in each). The data are represented as mean ± s.e.m. and all data were analyzed using a one-way ANOVA followed by the Tukey post hoc test for multigroup comparison. The blots shown are representative of *n* = 2 biological replicates. Source data are provided as a Source Data file.

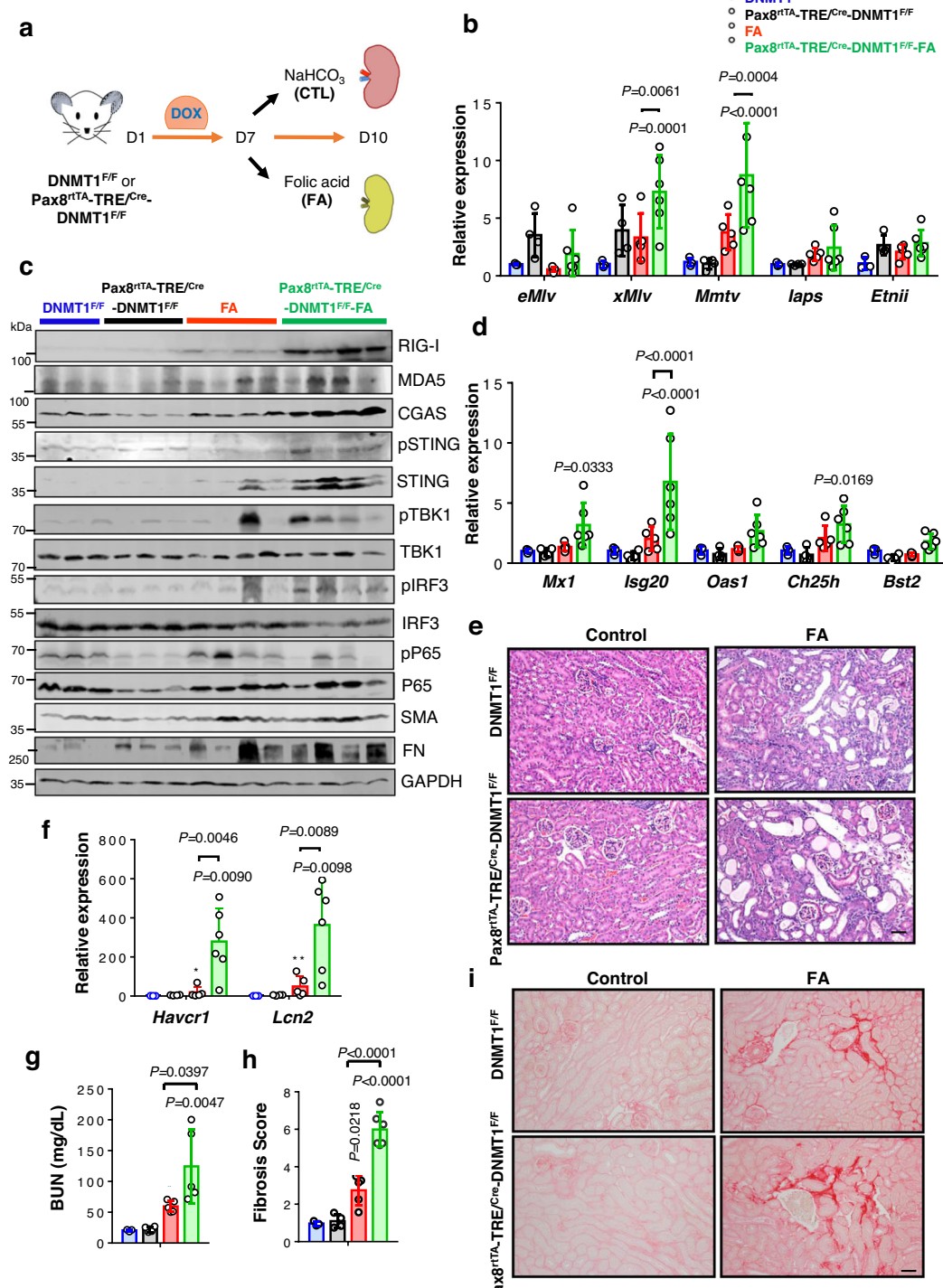

**Fig. 6 | Conditional inducible genetic deletion of DNMT1 in kidney tubule cells will lead to increased TE and ERV expression and more severe kidney injury.**
**a** Experimental design. **b** Relative mRNA levels of Class I ERVs (*eMlv, xMlv*, and *Mmtv*) and Class II ERVs (*Iaps* and *Etnii*) in kidneys of DNMT1^(F/F) and Pax8^(rtTA)/TRE^(Cre)/DNMT1^(F/F) mice injected with FA or vehicle (DNMT1^(F/F) (blue), *n* = 3; Pax8^(rtTA)/TRE^(Cre)/DNMT1^(F/F) (black), *n* = 4; FA (red), *n* = 5; Pax8^(rtTA)/TRE^(Cre)/DNMT1^(F/F) FA (green), *n* = 6). **c** Western blot showing representative images of RIG-I, cGAS, pSTING, STING, pTBK1, TBK1, pIRF3, IRF3, pP65, P65, SMA, and FN protein levels in kidneys of DNMT1^(F/F) and Pax8^(rtTA)/TRE^(Cre)/DNMT1^(F/F) mice injected with FA or vehicle. GAPDH was used as a loading control. **d** Relative mRNA levels of ISGs (*Mx1, Isg20, Oas1, Ch25h*, and *Bst2*) in kidneys of DNMT1^(F/F) and Pax8^(rtTA)/TRE^(Cre)/DNMT1^(F/F) mice injected with FA or vehicle. (DNMT1^(F/F), *n* = 3; Pax8^(rtTA)/TRE^(Cre)/DNMT1^(F/F), *n* = 4; FA, *n* = 5; Pax8^(rtTA)/TRE^(Cre)/DNMT1^(F/F) FA, *n* = 6). **e** Representative hematoxylin and eosin (H&E) staining of kidneys from DNMT1^(F/F), Pax8^(rtTA)/TRE^(Cre)/DNMT1^(F/F) mice treated with FA

or sham. Scale bar, 10 μm. **f** Relative mRNA levels of kidney injury markers (*Havcr1* and *Lcn2*) in kidneys of DNMT1^(F/F) and Pax8^(rtTA)/TRE^(Cre)/DNMT1^(F/F) mice injected with FA or vehicle (DNMT1^(F/F), *n* = 3; Pax8^(rtTA)/TRE^(Cre)/DNMT1^(F/F), *n* = 4; FA, *n* = 5; Pax8^(rtTA)/TRE^(Cre)/DNMT1^(F/F) FA, *n* = 6). **g** Blood urea nitrogen (BUN) levels of DNMT1^(F/F) and Pax8^(rtTA)/TRE^(Cre)/DNMT1^(F/F) mice injected with vehicle or FA (DNMT1^(F/F), *n* = 3; Pax8^(rtTA)/TRE^(Cre)/DNMT1^(F/F), *n* = 4; FA, *n* = 5; Pax8^(rtTA)/TRE^(Cre)/DNMT1^(F/F) FA, *n* = 5). **h** The degree of fibrosis quantified by Sirius red staining of kidneys of DNMT1^(F/F) and Pax8^(rtTA)/TRE^(Cre)/DNMT1^(F/F) mice injected with vehicle or FA (DNMT1^(F/F), *n* = 3; Pax8^(rtTA)/TRE^(Cre)/DNMT1^(F/F), *n* = 4; FA, *n* = 5; Pax8^(rtTA)/TRE^(Cre)/DNMT1^(F/F) FA, *n* = 5). **i** Representative images of kidneys of Sirius red staining of DNMT1^(F/F) and Pax8^(rtTA)/TRE^(Cre)/DNMT1^(F/F) mice injected with vehicle or FA. Scale bar, 10 μm. The data were represented as mean ± s.e.m. and all data were analyzed using a one-way ANOVA followed by Tukey post hoc test for multigroup comparison (**b, d–h**). Data were representative of two independent experiments (**c, e, i**). Source data are provided as a Source Data file.

and the expression of some of them were even higher (*xMlv* and *Mmtv*) in Pax8$^{rtTA}$-TRE/$^{Cre}$-DNMT1$^{F/F}$-FA and -UUO mice compared to DNMT1$^{F/F}$-FA and DNMT1$^{F/F}$-UUO, respectively (Fig. 6a and Fig. S11b). Expression of cytosolic nucleotide sensors *Rigi, Mda5, Tlr7, Cgas*, and *Sting* (Fig. 6c and Figs. S10b, S11c) and their downstream targets, TBK1 and IRF3 were higher in Pax8$^{rtTA}$-TRE/$^{Cre}$-DNMT1$^{F/F}$-FA mice (Fig. 6c). Transcript levels of type I IFN genes (*Ifnb, Ifne*, and *Ifnk)* (Fig. S10c) and ISGs (*Mx1, Isg20, Oas1, Ch25h*, and *Bst2*) (Fig. 6d and Fig. S11d), inflammatory markers *Tnfa, Il6, Cxcl10*, and *Ccl2* (Fig. S10d) and immune cell markers *Lyz2* and *Cd68* (Fig. S10e) were markedly higher in FA-injected Pax8$^{rtTA}$-TRE/$^{Cre}$-DNMT1$^{F/F}$ mice compared to FA-injected DNMT1$^{F/F}$ mice and in Pax8$^{rtTA}$-TRE/$^{Cre}$-DNMT1$^{F/F}$-UUO mice compared to DNMT1$^{F/F}$-UUO mice. In addition, Pax8$^{rtTA}$-TRE/$^{Cre}$-DNMT1$^{F/F}$ mice injected with FA were found be more susceptible towards renal injury and fibrosis compared to FA-injected DNMT1$^{F/F}$ mice (Fig. 6e–i). We found similar phenotype in Pax8$^{rtTA}$-TRE/$^{Cre}$-DNMT1$^{F/F}$ mice following UUO surgery compared to DNMT1$^{F/F}$-UUO (Fig. S11e). Expression of pro-fibrotic genes such as *Tgfb1, Col1, Col3, Vim*, and *Fn* were higher in FA-injected Pax8$^{rtTA}$-TRE/$^{Cre}$-DNMT1$^{F/F}$ mice when compared to DNMT1$^{F/F}$ mice injected with FA (Fig. S10f). These findings suggest the role of DNMT1 loss-mediated ERV induction in renal inflammation and kidney injury.

### RIG-I and STING are important mediators of renal fibroinflammation

To define the role of potential ERV-mediated RIG-I activation in mouse CKD models, we analyzed RIG-I knock-out mice following kidney injury (Fig. 7a, Figs. S12 and S13a). We observed an increase in ERV RNA levels: Class I ERVs (*eMlv, xMlv, pMlv*, and *Mmtv*) and Class II ERVs (*Gln* and *EtnII*), in kidneys of FA-induced injury model (Fig. 7b). We found lower expression levels of cytosolic sensors *Rigi, Mda5, Tlr4, Tlr7, Cgas, Sting*, and *Aim2* in RIG KO FA and RIG KO-UUO kidneys compared to WT FA and WT UUO, respectively (Figs. S12a, S13b). RIG-I downstream targets including TBK1, IRF3/7, and NF-κB signaling pathway also showed lower expression in RIG KO FA kidneys compared to the WT FA model of kidney fibrosis (Fig. 7c). Type I IFN genes (*Ifnb, Ifne, Ifnk, Ifnar1*, and *Ifnar2)* (Figs. S12b, S13c), ISGs (*Mx1, Isg15, Ifitm1, Oas1, Ch25h*, and *Bst2*) (Fig. 7d and Fig. S13d), inflammatory markers such as *Tnfa, Il1b, Ccl2*, and *Cxcl10* (Fig. S12c) and immune cell markers *Lyz2* and *Cd68* (Fig. S12d) were markedly lower in FA-injected RIG KO mice. We validated the lower CD4, CD8, and macrophage cell fractions in RIG KO-UUO kidneys by IHC compared to WT-UUO mice (Fig. S4a). In addition, RIG KO mice injected with FA showed lesser renal injury and fibrosis compared to FA-injected WT mice (Fig. 7e–h). Expression of pro-fibrotic markers including *Tgfb1, Col1, Col3, Vim*, and *Fn* were also lower in RIG KO mice as compared to WT mice injected with FA (Fig. S12e). We found a similar protective phenotype in RIG KO-UUO mice compared to WT-UUO (Fig. S13d–h)[42]. These findings strongly suggest that RIG-I is an important mediator of renal inflammation in fibrosis.

As we observed only a partial protective phenotype in the RIG-I KO mice, we reasoned that other nucleotide sensors, such as STING might also contribute to disease development. We, therefore, generated double KO mice with RIG and STING deficiency to assess renal phenotype following FA-induced injury. As expected, *Rigi* and *Sting* expression was low in double KO mice (Fig. 8a). Double KO mice also showed lower expression of other cytosolic nucleic acid sensors (*Cgas* and *Mda5*) (Fig. 8b) and ISGs (*Mx1, Isg15, Oas1, Ch25h*, and *Bst2*) (Fig. 8c) compared to WT mice injected with FA. To further evaluate innate immune response, we isolated immune cells such as macrophages, dendritic cells (DC), natural killer (NK) cells, and CD8T cells from SHAM and UUO kidneys of WT and RIG KO STING KO mice. Proinflammatory cytokines such as *Il1b, Mcp1*, and *iNos* in macrophages; *Il1b* and *Il6* in DCs; *Il6, Il17*, and *IL22* in NK cells; and *Tnfa, Il6*, and *Il22* in CD8T cells were found lower in RIG KO STING KO-UUO mice

compared to WT-UUO mice. Except *Tgfb1* and *Il10*, RNA levels of other anti-inflammatory cytokines showed minimal changes in isolated immune cells (Fig. S14a–d). Single-cell gene expression data from control and UUO mice further confirmed the immune cell activation (Fig. S14e). Most importantly, these mice showed improved kidney function with reduced kidney injury and fibrosis following FA injection when compared to WT mice (Fig. 8d–h). Collectively, our data indicate an important role for the RIG-STING signaling in ERV-mediated renal fibroinflammation.

### Reverse transcriptase inhibitors alleviate fibroinflammation in mice

The RLR plays a role in recognizing RNA while cGAS/STING detects DNA. We observed that deletion of RIG-I or STING ameliorated kidney disease, we hypothesized that ERVs are actively transcribed to DNA by reverse transcriptase enzymes. Reverse transcriptase activity can be efficiently inhibited by FDA-approved antiretroviral drugs, for example, lamivudine (3TC, a nucleoside reverse transcriptase inhibitor). We, therefore, induced kidney fibrosis by FA injection and treated mice with vehicle or 3TC (Fig. 9a). We found that 3TC-treated mice had lower ERV DNA levels compared to vehicle-treated mice (Fig. 9b). We found reduced RNA and protein expression of RIG-I, MDA5, and STING in 3TC-treated FA mice kidneys (Fig. 9c and Fig. S15a). TBK1-mediated activation of IRF3 was also dampened in 3TC-treated FA mice kidneys when compared to sham-treated FA mice (Fig. 9c). Expression of ISGs (*Mx1, Isg15, Isg20, Ifitm1, Bst2*, and *Oas1*) (Fig. 9d), and immune cell markers *Lyz2* and *Cd68* were markedly lower in 3TC mice with FA-induced kidney injury compared to controls (Fig. S15b). We found that 3TC-treated mice showed improved kidney function (Fig. 9e) with lesser kidney injury and fibrosis after FA injection as compared to vehicle control (Fig. 9f–h). Furthermore, transcript levels of pro-fibrotic genes such as *Tgfb1, Col1, Col3, Vim*, and *Fn* (Fig. S15c) were markedly lower in 3TC mice with FA-induced kidney injury compared to control. These results suggest that reverse transcriptase inhibitor treatment can alleviate ERV-mediated activation of innate immune response and inflammation.

Next, we wanted to understand whether the reverse transcriptase treatment can also protect kidneys from UUO-induced kidney fibrosis (Fig. 10a). We found that 3TC-treated mice showed lower levels of ERV DNA compared to untreated mice (Fig. 10b). RNA and protein levels of cytosolic nucleic acid sensors (RIG-I, MDA5, CGAS, and STING) were low in UUO mice treated with 3TC compared to mock-treated UUO mice (Fig. 10c and Fig. S16a). Downstream signaling molecules in the RIG and STING pathway such as phosphorylation of TBK1 and IRF3 were markedly lower in 3TC-UUO mice compared to CTL-UUO animals (Fig. 10c). We found that expression of ISGs (*Mx1, Isg15, Isg20, Ifitm1, Bst2*, and *Oas1*) (Fig. 10d), markers of inflammation such as *Tnfa, Il1b, Il6, Ccl2*, and *Cxcl10*, and immune cell Including *Lyz2* and *Cd68* were lower in 3TC mice with UUO compared to CTL-UUO (Fig. S16b). We also found markedly lower macrophages, CD4 T cell, and CD8 T lymphocytes in 3TC-treated UUO mice compared to sham-treated UUO mice (Fig. S4a). Mice treated with 3TC showed less renal injury and fibrosis (Fig. 10e–g). RNA levels of pro-fibrotic genes such as *Tgfb1, Col1, Col3, Vim*, and *Fn* were also lower in kidneys from 3TC-UUO compared to CTL-UUO (Fig. S16c). These results indicated that reverse transcriptase inhibitors can attenuate IFN response and kidney fibrosis.

### Discussion

Here, we show an increase in ERV expression in human diabetic and hypertensive CKD, their role in activating the RIG-I, STING pathways, and their contribution to fibroinflammation. By analyzing more than 240 human kidney samples and several mouse CKD models, we

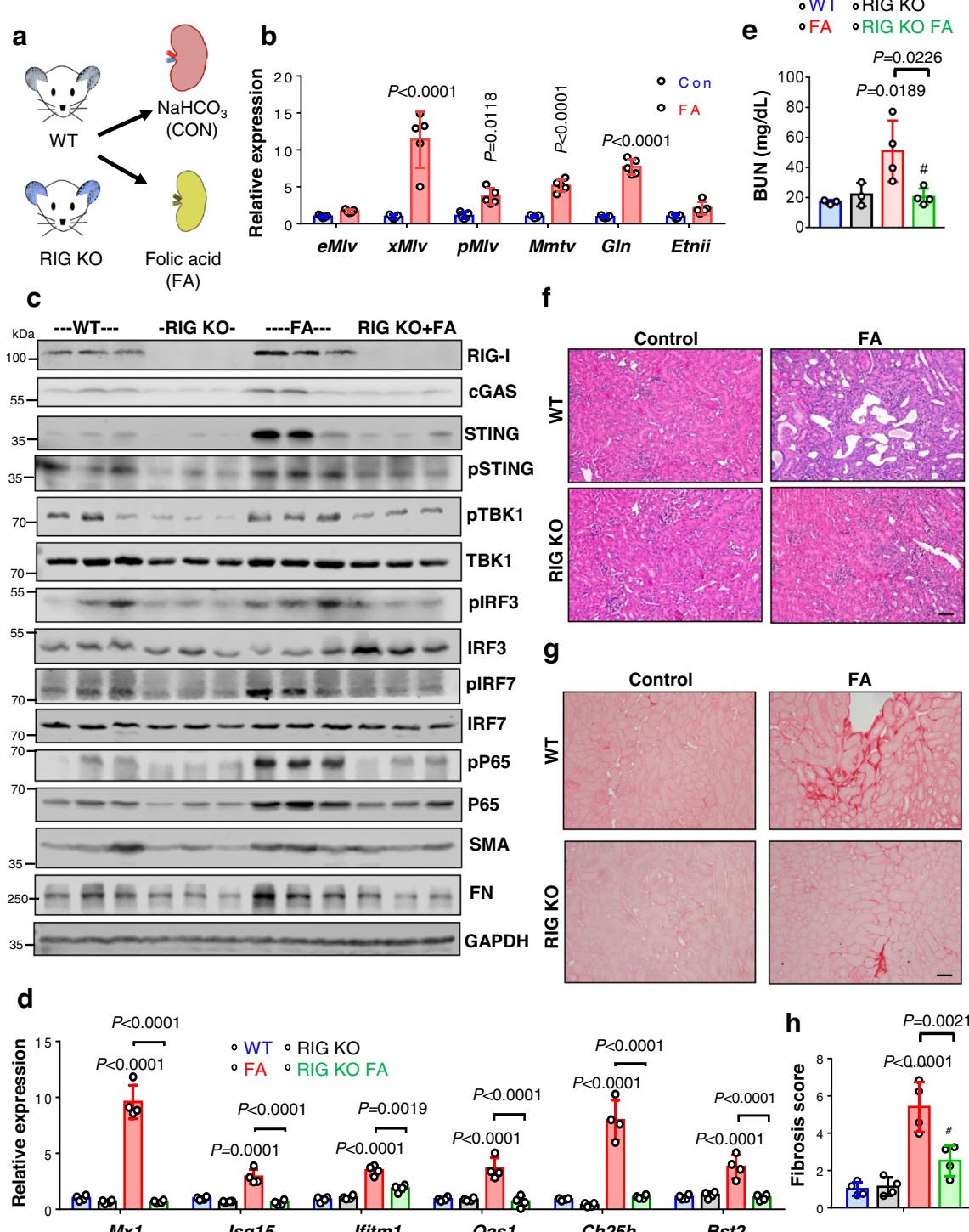

**Fig. 7 | RIG-I deficiency ameliorates FA-induced kidney fibrosis. a** Experimental design: WT and RIG KO mice were injected with FA or vehicle and animals were sacrificed on day 7. **b** Relative mRNA levels of Class I ERVs (*eMlv, xMlv, pMlv*, and *Mmtv*) and Class II ERVs (*Gln* and *EtnII*) in kidneys of WT mice injected with vehicle (blue) or FA (red) (*n* = 5 in each). Data were presented as the mean ± s.e.m and analyzed by two-tailed unpaired Student's *t*-test. **c** Western blots showing representative images of RIG-I, cGAS, pSTING, STING, pTBK1, TBK1, pIRF3, IRF3, pIRF7 and IRF7, pP65, P65, SMA, and FN protein levels in kidneys of WT and RIG KO mice injected with vehicle or FA. GAPDH was used as a loading control. **d** Relative mRNA levels of ISGs (*Mx1, Isg15, Ifitm1, Oas1, Ch25h*, and *Bst2*) in kidneys of WT and RIG KO mice injected with vehicle or FA (*n* = 4 in each). **e** Blood urea nitrogen (BUN) levels of WT and RIG KO mice injected with vehicle or FA (WT (blue) and RIG KO (black), *n* = 3 in each; FA (red) and RIG KO FA (green), *n* = 4 in each). **f** Representative hematoxylin and eosin (H&E) staining of kidneys from WT and RIG KO mice injected with vehicle or FA. Scale bar, 10 μm. **g** Representative images of Sirius red staining of WT and RIG KO mice injected with vehicle or FA kidneys. Scale bar, 10 μm. **h** The degree of fibrosis quantified by Sirius red staining of WT and RIG KO mice injected with vehicle or FA kidneys (*n* = 4 in each). The data were represented as mean ± s.e.m. and all data were analyzed using a one-way ANOVA followed by Tukey post hoc test for multigroup comparison (**d**, **e**, **h**). Data were representative of two independent experiments (**c**, **f**, **g**). Source data are provided as a Source Data file.

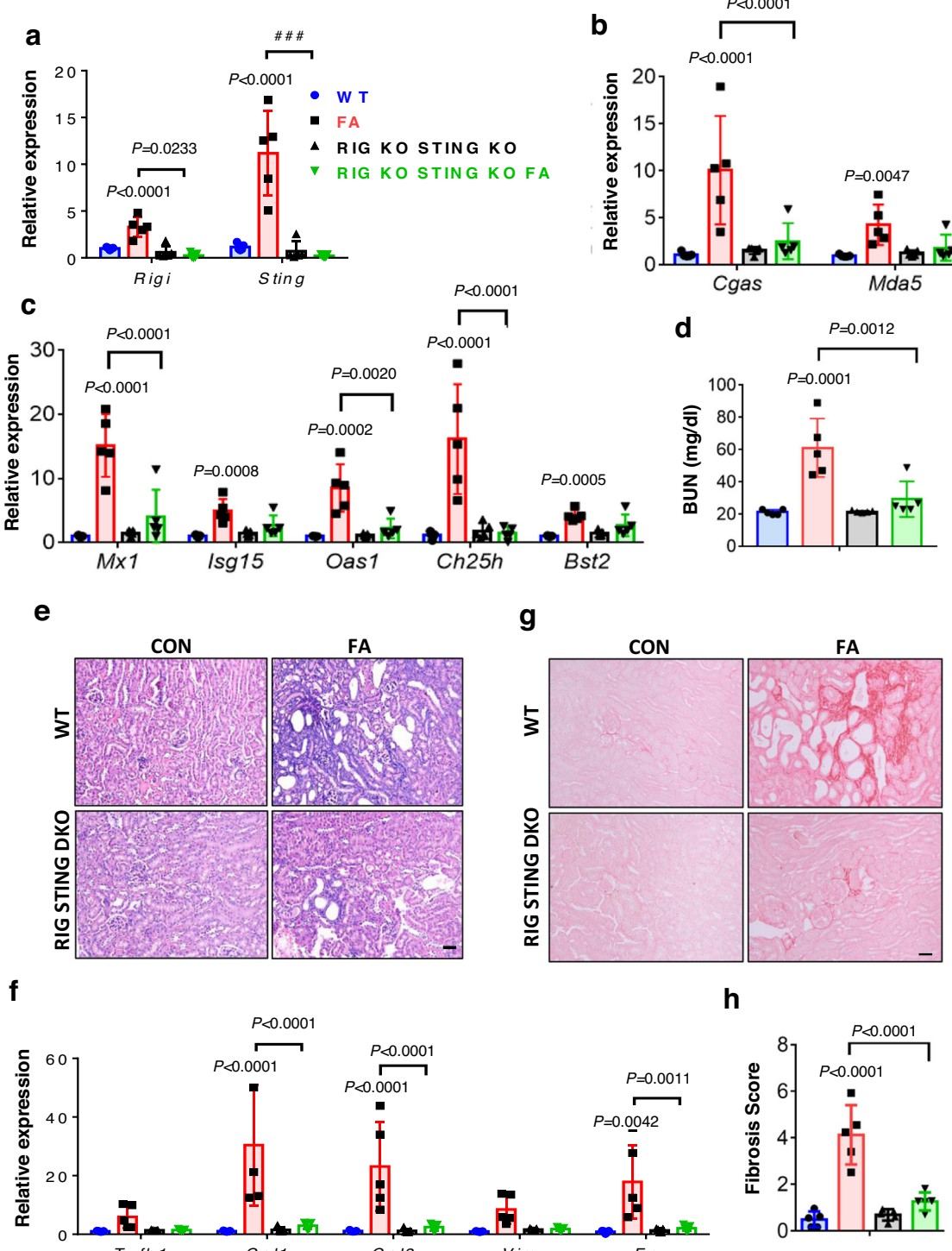

**Fig. 8 | RIG KO STING KO (Double KO) mice were protected from FA-induced injury. a** Relative mRNA levels of *Rigi* and *Sting* in kidneys of WT (vehicle, blue; FA, red) and RIG KO STING KO mice (vehicle, black; FA, green) injected with FA (*n* = 5 in each). **b** Relative RNA levels of other cytosolic RNA sensors (*Cgas*, and *Mda5*) in kidneys of WT and RIG KO STING KO mice injected with FA (*n* = 5 in each). **c** Relative mRNA level of ISGs (*Mx1, Isg15, Oas1, Ch25h,* and *Bst2*) in kidneys of WT and RIG STING KO mice injected with FA (*n* = 5 in each). **d** Serum BUN levels of WT and RIG KO STING KO mice injected with FA (*n* = 5 in each). **e** Representative images of H&E staining of kidney sections from WT and RIG KO STING KO mice injected with FA.

Scale bar: 10 μm. **f** Relative mRNA level of fibrosis markers (*Tgfb1, Col1, Col3, Vim,* and *Fn*) in kidneys of WT and RIG KO STING KO mice injected with FA (*n* = 5 in each). **g** Representative images of Sirius red staining of kidney sections from WT and RIG KO STING KO mice injected with FA. Scale bar: 10 μm. **h** Quantification of tubulointerstitial fibrosis by Sirius red staining in WT and RIG KO STING KO mice injected with FA (*n* = 5 in each). The data are represented as mean ± s.e.m. and all data were analyzed using a one-way ANOVA followed by Tukey post hoc test for multigroup comparison (**a**–**d**, **f**, **h**). Data were representative of two independent experiments (**e, g**). Source data are provided as a Source Data file.

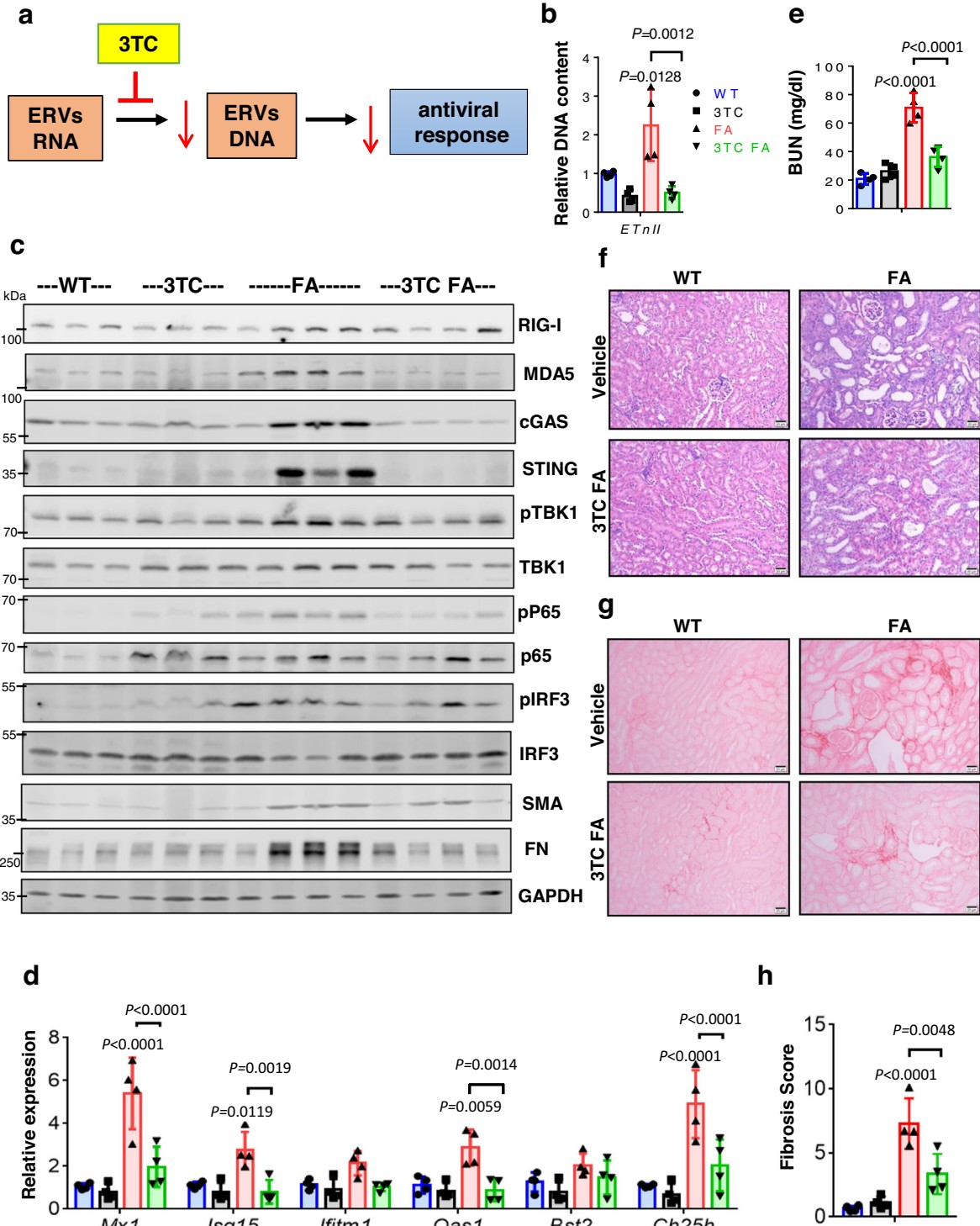

**Fig. 9 | Antiviral Lamivudine (3TC) treatment protected mice from FA-induced injury. a** Experiment design. **b** ERV DNA content (Etnii) in kidneys of WT and FA mice treated with or without 3TC (WT, blue; 3TC, black; FA, red; 3TC FA, green; $n = 4$ in each). **c** Western blots showing protein levels of RIG-I, MDA5, cGAS, STING, pTBK1, TBK1, pP65, P65, pIRF3, IRF3, FN, and SMA in kidneys of WT and FA mice treated with or without 3TC. GAPDH was used as a loading control. **d** Relative mRNA level of ISGs (*Mx1, Isg15, Ifitm1, Oas1, Bst2*, and *Ch25h*) in kidneys of WT and FA mice treated with or without 3TC ($n = 4$ in each). **e** Serum BUN levels of WT and FA mice treated with or without 3TC ($n = 4$ in each). **f** Representative images of H&E staining of kidney sections from WT and FA mice treated with or without 3TC. Scale bar: 10 μm. **g** Representative images of Sirius red stained kidney sections of WT and FA mice treated with or without 3TC. Scale bar: 10 μm. **h** Quantification of tubulointerstitial fibrosis by Sirius red staining in WT and FA mice treated with or without 3TC ($n = 4$ in each). The data are represented as mean ± s.e.m. and all data were analyzed using a one-way ANOVA followed by Tukey post hoc test for multigroup comparison (**b, d, e, h**). Data were representative of two independent experiments (**f, g**). The blots shown are representative of $n = 2$ biological replicates. Source data are provided as a Source Data file.

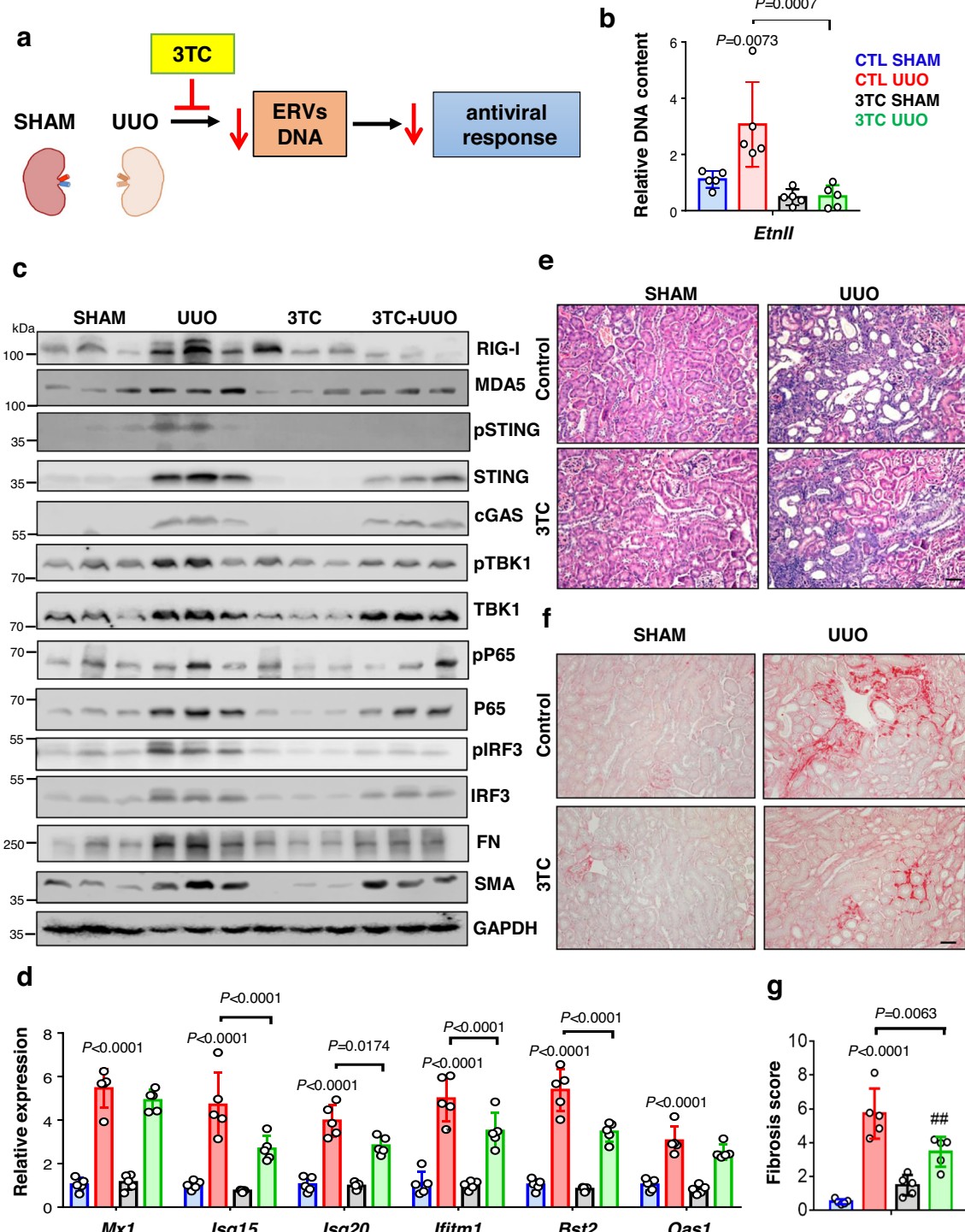

**Fig. 10 | The reverse transcriptase inhibitor Lamivudine ameliorates fibroinflammation. a** Experimental scheme. **b** Relative DNA content of Etnii in kidneys of mice underwent sham or UUO surgery and injected with vehicle or 3TC (CTL SHAM, blue; CTL UUO, red; 3TC SHAM, black; 3TC UUO, green; $n = 5$ in each). **c** Representative images of Western blots of RIG-I, MDA5, cGAS, pSTING, STING, pTBK1, TBK1, pP65, P65, pIRF3, IRF3, FN, and SMA protein levels in kidneys of sham and UUO mice injected with vehicle or 3TC. GAPDH was used as a loading control. **d** Relative mRNA levels of ISGs (*Mx1, Isg15, Ifitm1, Bst2,* and *Oas1*) in kidneys of SHAM and UUO mice injected with vehicle or 3TC ($n = 5$ in each). **e** Representative hematoxylin and eosin (H&E) staining of kidneys from sham and

UUO mice injected with vehicle or 3TC. Scale bar, 10 μm. **f** Representative images of Sirius red staining of kidneys from sham and UUO mice injected with vehicle or 3TC. Scale bar, 10 μm. **g** The degree of fibrosis score was quantified by Sirius red staining of kidneys from sham and UUO mice injected with vehicle or 3TC ($n = 5$ in each). The data are represented as mean ± s.e.m. and all data were analyzed using a one-way ANOVA followed by Tukey post hoc test for multigroup comparison (**b, d, g**). Data were representative of two independent experiments (**e, f**). The blots shown are representative of $n = 2$ biological replicates. Source data are provided as a Source Data file.

provide a comprehensive catalog of TEs and ERVs and their correlation with disease severity. Our studies indicate the role of decreased cytosine methylation in allowing TEs/ERVs expression in disease. TE and ERV levels correlated and lead to the activation of the cytosolic nucleotide sensing pathways, such as RIG-I/STING, leading to type I IFN response and ISGs expression. Loss of RIG-I or STING can alleviate ERVs induced inflammation in PTECs and mice kidneys. RIG-I emerged as a key immunoregulatory factor and a potential therapeutic target that plays a crucial role in renal inflammation and CKD progression.

In our survey, we believe that this is the first report to catalog TEs and ERVs expression in non-cancerous kidneys and show the correlation and increase in expression of ERVs and kidney fibrosis. Using RepeatMasker, we were able to identify all TE types such as ERVs, LINE, and SINEs wherein ERVs were found to be prominent TE in our dataset. Interestingly, differential expression analysis revealed that a greater fraction of TE showed a positive correlation with kidney fibrosis. We also analyzed full-length ERV sequences using HervQuant and found a similar increase in ERVs level which is comparable to previous reports[34,43]. We found a similar pattern by analyzing mouse kidney fibrosis samples, such as a greater number of TEs showed an increase in expression in fibrosis. Due to differences in genomic structure and ERVs integration, it is difficult to compare the human ERVs with murine ERVs at individual levels. However, Class I and Class II ERVs were the most abundant ERVs in both species.

Our results indicate that loss of cytosine methylation and epigenetic derepression likely plays an important role in ERV activation. In human kidney samples, we identified loss of methylation was associated with higher ERV levels. Guo et al. earlier suggested a protective role of DNA methylation in cisplatin-induced kidney injury[44], however, the mechanism of DNMT1-afforded protection was not fully explained. DNMT1 loss in nephron progenitor cells (Six2+ cell) caused a marked loss of methylation of transposable elements and severe kidney developmental damage[45,46]. Loss of different epigenetic regulators causes severe developmental defects in the kidney[47–50] with higher inflammatory gene signatures suggesting that ERVs are potent immunostimulants. We found that mice with conditional inducible deletion of DNMT1 in tubule cells were more susceptible to kidney injury and showed higher inflammation and fibrosis, indicating the role of DNMT1 in the adult and not just in developing kidneys.

It is important to note that loss of cytosine methylation is an important feature of both kidney fibrosis[51] and aging. Both conditions are associated with enhanced inflammation. However, the mechanism of aging-induced inflammation is not well understood. Epigenetic derepression-induced immune activation has been shown to play a key role in the anti-cancer action of DNMT1i. Epigenetic derepression associated with ERV activation and enhanced inflammation might play an important role in clearing cancerous cells in the context of aging and fibrosis. We can't exclude the role of telomere reverse transcriptase[39] or other epigenetic regulators that might also regulate TE/ERVs expression. Further studies are needed to understand the role of epigenetic regulators such as SETDB1 and TRIM28 in regulating ERV levels in the kidney[52,53].

Viral RNA/DNA in the cytosol can activate innate immune pathways via cytosolic nucleic acid sensors. The most widely studied RNA sensors are RLRs (RIG-I and MDA5) and TLRs (TLR3/4 and TLR7), and DNA sensors: cGAS, STING, and AIM2 that lead to enhanced IFN response and cytokine release upon activation via TBK1 and IRF3/7. Transfection of full-length HERV-K, HERV RNA and dsRNA in cultured kidney tubule cells lead to the activation of RLRs (RIG-I and MDA5) and STING, while no change was observed in TLRs levels. As TLRs are mostly expressed by professional immune cells such as macrophages and dendritic cells[54], we, therefore, focused our attention on RLRs and STING signaling pathways in PT cells. We found activation of a downstream target of RIG-I and STING i.e. TBK1 and IRF3/7 leading to IFNs and ISGs

expression. We observed similar results when cells were treated with DNMT1 inhibitor AZA. We also observed increased levels of RIG-I, MDA5, cGAS, and STING proteins in diseased kidney tubules of patients and mice. Furthermore, the expression of STING and RLRs strongly correlated with ERV levels suggesting the role of ERV in activating RIG-I and STING. A similar role for ERV have been proposed in neurodegenerative disease, cancer, and autoimmune diseases such as SLE[55–57].

Deletion of STING or/and RIG-I protected mice from fibrosis development in the folic acid-induced disease model, indicating the key role of these pathways in kidney fibroinflammation. Further studies shall analyze the role of MDA5, AIM2, and MAVS. PT cells have been reported as a main producer of IFNα[58,59] and showed strong type I IFN-induced response upon polyomavirus infection[59]. We found *Ifnk* is a type of IFN released from PT cells during injury. Its receptor; IFNAR1/2 was highly expressed in immune cells such as macrophages suggesting crosstalk between these two cell types. Previous studies have shown IFNβ and IFNAR1/2 blockade improved kidney function and histology in kidney disease models suggesting the importance of IFN signaling in kidney disease[60–62]. So, it will be important to further dissect the role of IFN signaling in CKD in future studies.

Finally, our work has important potential therapeutic implications such as the use of STING and RIG-I inhibitors in kidney fibroinflammation. In addition, the reverse transcriptase inhibitor lamivudine also ameliorated kidney damage in mouse fibrosis models. Given the low toxicity of 3TC, it would be interesting to examine its effect on patients. In addition, as AZA is being used in a variety of cancer types, looking into renal and other organ inflammation and fibrosis will be important.

In summary, here we present a comprehensive TE/ERV analysis of healthy and diseased kidney samples. We show that the epigenetic derepression-mediated increase in ERVs expression activates the nucleotide sensing pathways (RIG-I/STING) inducing a sterile inflammation and fibrosis. Our study raises the possibility of using STING, RIG-I inhibitors, and NRTI for the treatment of renal fibroinflammation.

## Methods
### Human samples
Deidentified human kidney samples were obtained from the non-neoplastic portion of surgical nephrectomies via the Cooperative Human Tissue Network approved by the University of Pennsylvania Institutional Review Board. Laboratory data (including serum creatinine) and demographic and clinical information including age, sex, self-reported ethnicity, diabetes, and hypertension status was collected from medical records by an honest broker, therefore informed consent was not required because the samples were permanently deidentified and no PHI (protected health information) information was collected. The Human study was deemed exempt by the institutional review board (IRB) of the University of Pennsylvania (exemption IV involves analysis of existing, deidentified, or publicly available data, please refer to Page 6 https://irb.upenn.edu/sites/default/files/Minimal%20Risk %20Research%20IRB%20FAQ.pdf). Therefore, consent was not necessary. The primary cohort included 240 (for TE analysis) and 485 (for full-length ERVs analysis) human kidney samples that were collected from the unaffected portion of surgical nephrectomies via the CHTN network. The collected kidney samples were immersed into RNAlater and stored at −80 °C. Tubule compartments were manually microdissected from the tissue and DNA and RNA were isolated. CKD diagnosis was established using histopathological analysis. Hypertension and diabetes diagnosis were based on chart review diagnosis codes. Blood pressure values were obtained at the time of tissue procurement.

Estimated GFR was determined using the Chronic Kidney Disease Epidemiology Collaboration[1] estimation equation.

## Mouse models

All animal experiments were reviewed and approved by the Institutional Animal Care and Use Committee of the University of Pennsylvania and were performed in accordance with the institutional guidelines (protocol #804138). Mice were housed in a pathogen-free animal house (12 h dark/light cycles) in a temperature- and humidity-controlled environment (23 ± 1 °C) and fed with a standard mouse diet and water ad libitum. 6 to 8-week-old male/female C57BL/6 wild-type mice were used in the study (Jackson Laboratories, stock #000664). RIG KO mice were purchased from Jackson Lab (Stock#46070). STING KO mice were purchased from Jackson Lab (Stock#025805). RIG KO STING KO (Double KO) mice generated by breeding RIG KO and STING KO mice. Mice were injected with FA (Fisher Scientific, #AC216630500) (250 mg/kg once, dissolved in 300 mM NaHCO$_3$) intraperitoneally and sacrificed on day 7 or mentioned otherwise. For the unilateral ureteral obstruction model, mice underwent ligation of the left ureter and were sacrificed on day 7. DNMT1 F/F mice were obtained from Mutant Mouse Regional Resource Center (MMRRC_014114-UCD) were crossed with TetO-Cre (TRE$^{Cre}$) (Jackson Laboratory, stock #006234) and Pax8$^{rtTA}$ mice (Jackson Laboratory, stock#007176) to generate Pax8$^{rtTA}$/TRE$^{Cre}$/DNMT1$^{F/F}$ mice. Pax8$^{rtTA}$/TRE$^{Cre}$/DNMT1$^{F/F}$ mice were fed on Dox-diet purchased from Bio-Serv (cat #S3888) for 7 days to knock-down DNMT1 expression and were maintained under the same conditions as WT mice. The number of animals in each group is specified in each figure legend. Six to eight weeks old (both male and female) mice were used in all the experiments. The efficacy study of antiviral 3TC inhibitor in FA and UUO WT mice was conducted as follows: WT mice (8–10 weeks old) were injected with lamivudine (3TC) 50 mg/kg body weight via i.p. (TCI America, L0217)[63] or PBS once per day starting 2 days prior to UUO surgery or FA injection and continued till day 7 post-UUO surgery and FA injection. For an anti-IFNG experiment, WT mice (8 weeks old) were injected with 1 mg/mouse IFNG antibody (XMG1.2, BioXcell, Cat#BE0055) on day −1 and day 3 via i.p. Mice were injected with FA on day 0 and sacrificed on day 7. All mice were euthanized at either endpoint or earlier as noted in the experiments. All mice were euthanized using CO$_2$ air displacement followed by cervical dislocation at time points as noted in the experiments.

## RNA Sequencing analysis

Human kidney RNA was isolated using the RNeasy Mini kit (QIAGEN). Sequence quality was first surveyed with FastQC and Trim Galore was used to trim low-quality and adapter sequences[64]. Trimmed reads were aligned to the human genome[43] using STAR-2.4.1d[65]. The aligned reads were mapped to RepeatMasker annotation and TEs were quantified using HTSeq-0.12.4[66]. Principal components analysis was performed on the samples to identify and remove all outliers (n = 15) (Fig. S1h). In addition, only TEs with a minimal expression of 1 TPM in at least 20% of samples were used for the analysis. A total of 106,540 TEs were detected, and 240 samples were analyzed. A linear regression model was implemented to examine the association between TE expression and interstitial fibrosis, using age, gender, race, diabetes, hypertension, batch, RIN, duplication, mitochondrial percentage, unmapped reads, and unique reads as covariates. A Benjamini–Hochberg (FDR) adjusted significance threshold of 0.05 was considered statistically significant. In silico deconvolution analysis of RNA-seq data was performed using CIBERSORT[40] using single-cell gene expression data from mouse kidney

samples[41,67] Pearson's correlation coefficients were calculated between TE expression and twenty-five cell types including endothelial cells (Endo), podocytes (Podo), proximal tubule cells (PT), loop of Henle (LOH), distal convoluted tubule (DCT), principal cells (PC), intercalated cells (IC), collecting duct (CD), macrophages (Macro), monocytes (Mono), B cells (B), granulocytes (Granul), CD cells (CD8T, CD8 effector, CD4T), Th-17 cells, T-reg cells, and immune cells (Immune).

HervQuant was utilized to quantify the expression of full-length ERV proviruses from the trimmed reads[34]. Principal components analysis was performed on the samples to identify and remove all outliers (Fig. S1i). Only ERVs with a minimal expression of 1 TPM in at least 20% of samples were used for the analysis. Following processing, a total of 2600 full-length ERVs were detected, and 485 samples remained. A linear regression model was implemented to examine the association between ERV provirus expression and interstitial fibrosis, adjusted for age, gender, race, diabetes, hypertension, batch, RIN, duplication, mitochondrial percentage, unmapped reads, and unique reads. Benjamini–Hochberg (FDR) adjusted significance threshold of 0.05 was considered statistically significant.

Mouse RNA was isolated using the RNeasy Mini kit (QIAGEN). A total of 200 ng RNA was used to isolate poly A purified mRNA using the Illumina TruSeq RNA Preparation kit. Illumina was used to perform high-throughput sequencing with 150-bp paired-end sequencing according to the manufacturer's instructions. Sequence quality was first surveyed with FastQC and Trim Galore was used to trim low-quality and adapter sequences[64]. Trimmed reads were aligned to the Gencode mouse genome (GRCm38) using STAR-2.7.8a[65]. To quantify gene expression, the aligned reads were mapped to the mouse gene annotation (GRCm38; version 7 Ensembl 82) using RSEM-1.3.0[68]. To quantify TE expression, the aligned reads were mapped to the mouse RepeatMasker annotation and TEs were quantified using HTSeq-0.12.4[66]. DESeq2 was used to test for differential TE expression between control and FA/UUO groups.

## Illumina Infinium EPIC BeadChip analysis

Illumina Infinium EPIC methylation array was used to profile CpG methylation in 240 human kidney samples. R package SeSAMe[69] was used for quality control and methylation quantification. After filtering out probes with missing values in more than 20% of samples, non-CpGs, low mapping quantity, chromosome X, Y, and M, and methylation overlapping with SNPs, 728,582 CpGs remained for further analysis. CpGs overlapping with TE sequences were extracted using the R package Granges[70]. β values for each CpG probe represent the methylation level, ranging from 0 to 1, or unmethylated to methylated, respectively. A total of 10,660 CpG probes overlapping TE sequences were identified. 380 CpG probes overlap the 2,711 TE sequences associated with fibrosis from our cohort. Of which, 308 are from unique TE. Pearson's correlation coefficients between the methylation level of each CpG probe and the expression of the TE it overlaps was calculated.

## Primary cell culture isolation and HKC-8 cell line

Primary tubular epithelial cells (PTECs) were isolated from 3–4 weeks old WT, RIG KO, and STING KO mice kidneys. Briefly, kidneys were minced into pieces and digested in 10 ml RPMI medium (Gibco, #21875-034) containing 100 μl of Collagenase IV (1 mg/ml, Sigma Aldrich) for 30 min at 37 °C. Afterward, collagenase IV activity was stopped by adding 100 μl of fetal bovine serum (FBS). Cells were sequentially sieved through 100, 70, and 40 μm nylon mesh and centrifuged for 10 min at 400×g. The pellet was resuspended in 1 ml of sterile RBC lysis buffer and incubated for 3 min on ice. DPBS (~9 ml) was added followed by centrifugation for 10 min at 400×g. The pellet was then resuspended in PTECs media (RPMI 1640 supplemented with 10% FBS, 20 ng/ml hEGF, 20 ng/ml bFGF, 1X ITS (insulin-transferrin-

selenium), and 1% penicillin-streptomycin). HKC-8 cells were obtained from Dr. Lorraine C. Racusen (The Johns Hopkins University School of Medicine). HKC-8 cells were cultured in DMEM/F12 supplemented with 5% FBS, 1X ITS, 0.5 µg/ml, hydrocortisone, and 1% Penicillin-Streptomycin at 5% $CO_2$, 37 °C.

## PTECs plasmid transfection
PTECs were transfected with pcDNA3.1 (vector control) or pcDNA3.1-HERV-K full-length[28] (kind gift from Dr. Avindra Nath, National Institute of Neurological Disorders and Stroke) using Lipofectamine 3000. For transfection, cells were seeded in 6-well plates, grown overnight for 60–70% confluency, and then transfected with 5 µg of vector/HERV-K overexpression plasmid. Cells were harvested 24 h post-transfection.

## Immunoprecipitation experiments
For dsRNA IP, we used 500 µg of whole kidney lysate from SHAM and UUO mice and was incubated with 1 unit RNase I (Ambion, Waltham, MA, USA). IP was performed with 5 µg of J2 anti-dsRNA antibody (Scicons, Szirák, Hungary)[71] or IgG control pre-bound to 15 µl of Protein A/G Dynabeads (Invitrogen, Waltham, MA, USA) with end-to-end rotation at 4 °C for 2 h. Beads were recovered using a magnet and washed three times with 1× IP buffer (20 mM Tris-HCl pH 7.5, 0.15 M NaCl, 0.1 mM EDTA, 0.1% Tween20). The dsRNA was recovered from the beads by adding 150 µL 1× IP buffer and 450 µL of Trizol LS (Ambion, Waltham, MA, USA) and then following the manufacturer's protocol for isolating RNA. dsRNA was quantified with Nanodrop and normalized to kidney weight to calculate dsRNA content in SHAM and UUO kidneys.

For dsRNA interaction with RIG-I, SDS lysis buffer (CST, Cat# 7722) was added to dsRNA bound beads and followed by western blot with RIG-I antibody. For reverse CO-IP, 2 µg of RIG-I antibody or IgG control were incubated with 500 µg of whole kidney lysate from UUO mice. RNA content was recovered by adding 150 µL 1× IP buffer and 450 µL of Trizol to beads bound with RIG-I followed by RNA isolation.

## dsRNA transfection
PTECs were transfected with immunoprecipitated dsRNA from UUO samples using Lipofectamine 3000 (Thermo, #11668027). For transfection, cells were seeded in 6-well plates, grown overnight for 60–70% confluency, and then transfected with 1.5 µg of dsRNA or nonspecific RNA from IgG-control immunoprecipitation. Cells were harvested 48 h post-transfection.

## IFNβ treatment
PTECs were treated with mouse IFNβ (1000 U/ml) (Biotechne, #8234-MB) or PBS for 24 h followed by RNA isolation and qRT-PCR.

## AZA treatment
PTECs were treated daily with 0.2 µM 5'-Aza-2'deoxycytidine (Sigma, #A3656) or DMSO (Merck, #D2650) for 3 consecutive days. After 72 h, fresh media was added without AZA and cells were cultured for 2 additional days and harvested on day 5. Mock-treated cells were treated with DMSO for 3 days (days 0, 1, and 2), and RNA and protein were harvested on day 5.

## In vitro HERV RNA transfection
HERV sequences {HERV 1132 (Chr3:137600973 + 137601191), HERV 3110 (Chr9:93754223-93754424), HERV 4321 (Chr3:101698330 + 101698523), HERV 4329 (Chr15:101075557 + 101075744), and HERV 915 (Chr3: 34683313 + 34683476)} were selected based on our data and cloned in pcDNA3.1 vector using BamHI and NotI as cloning site. Empty pcDNA vector and HERV clones were further linearized by XhoI restriction enzyme and used as templates for in vitro RNA transcription using MEGAscript™ T7 Transcription Kit according to manufacturer guidelines (Thermo Scientific, Cat# AM1334). Purified HERV RNA were quantified and 2 µg was used to transfect PTECs and HKC-8

cells. Cells were harvested for RNA and protein isolation 12 h post-transfection.

## Immune cells isolation from kidney
Macrophages, dendritic cells, NK cells, and CD8T cells were isolated from SHAM and UUO single-cell kidneys suspension of WT and RIG KO STING KO mice. F4/80 microbeads were used for macrophages isolation (Miltenyi Biotec, Cat#130-110-443), EasySep™ Mouse CD11c Positive Selection Kit II was used for DC cells isolation (STEMCELL Technologies, Cat# 18780), NK isolation kit was used for NK cells isolation (Miltenyi Biotec, cat# 130-115-818), EasySep™ Mouse CD8 + T Cell Isolation Kit was used for CD8T cells isolation (STEMCELL Technologies, cat# 19853) according to manufacturer guidelines.

## RNA isolation and qRT-PCR
To isolate total RNA, kidney tissue samples or cells were homogenized in TRIZOL according to the manufacturer's instructions. Pellets were dried for 10–15 min and dissolved in nuclease-free water. RNA was treated with DNase I to remove DNA contamination. RNA (2 µg) was reverse-transcribed using a cDNA synthesis kit (Applied Biosystems, #4368813)[72]. The cDNA was stored at −20 °C until use. The qPCR was performed using SYBR green dye (Applied Biosystems, #4367659) under the ViiA 7 System (Life Technologies). The data were normalized and analyzed using the ddCT method. The primers used are listed in Supplementary Data S7.

## Protein extraction and western blotting
Tissue or cells were homogenized in SDS lysis buffer Cell Signaling Technology (CST), (cat#7722) and were separated by SDS-PAGE and then transferred to PVDF membranes. After blocking, for 30 min with 5% milk or 5% BSA (for phospho antibodies) in TBST, membranes were incubated overnight with primary antibody (HERV-K (1:100) (Novus, #H00002087-A01), RIG-I (1:1000) (CST, #3743), MDA5 (1:1000) (CST, #5321), STING (1:2000) (CST, #13647), pSTING (1:1000) (CST, #19781), cGAS (1:2000) (CST, #31659), pTBK1 (1:2000) (CST, #5483), TBK1 (1:2000) (CST, #3504), IRF3 (1:2000) (CST, #1190 T), pIRF3 (1:1000) (CST, #37829), GAPDH (1:5000) (CST, #2118), ACTIN (1:10,000) (Sigma, A3854), IRF7 (1:2000) (CST, #4920), pIRF7 (1:2000) (CST, #24129), pP65 (1:1000) (CST, #3033), P65 (1:2000) (CST, #8242), FN (1:2000) (Abcam, #ab2413), IKKε (1:1000) (CST, #3416), pIKKε (1:1000) (CST, #8766) and SMA (1:2000) (Sigma, #A5228) in TBST. After three washes for 5 min, membranes were incubated with horseradish peroxidase (HRP)–conjugated secondary anti-rabbit (1:5000) (CST, #7074) or anti-mouse antibody (1:5000) (CST, #7076) or IR-conjugated anti-mouse (1:20,000) (CST, #5470 S), and anti-rabbit (1:20,000) (CST, #5151S) antibody in TBST. The signal was developed with Immobilon forte western HRP substrate (Millipore, #WBKLS0500) using Odyssey®Fc Imaging System (LICOR) equipment and measured with Image Studio Lite software.

## In situ hybridization
In situ hybridization was performed using formalin-fixed paraffin-embedded tissue samples and the RNAscope 2.5 HD Duplex Detection Kit (ACD-Bio, #322436). We followed the manufacturer's original protocol. The HERV-K gag-env (cat#443391-C1) probe was used for the RNAscope assay.

## Histological analysis
Kidney samples were fixed in 10% neutral formalin and paraffin-embedded sections were stained with hematoxylin and eosin (H&E) to analyze the histology of samples. Sirius red staining (Boekel Scientific, #147122) was performed to determine the degree of fibrosis. ImageJ software was used to quantify the fibrosis score. We performed immunocyto- and histo-chemistry on paraformaldehyde-fixed cells and formalin-fixed, paraffin-embedded kidney sections. We used the

following dilutions of the primary antibodies: RIG-I (1:200), cGAS (1:200), STING (1:200), pSTING (1:200), HERV-K (1:100), CD4 (1:100) (Thermo, #36-0041-85 for mouse and #14-0049-82 for human), CD8 (1:100) (Thermo, #36-0081-85 for mouse, and #12-0088-80 for human), and F4/80 (1:200) (Thermo, #MA191124). Staining was visualized using peroxidase-conjugated antibodies anti-mouse or anti-rat antibodies using the Vectastain Elite kit (Vector Lab, #PK-6102 and #PK-6104) and 3,3-diaminobenzidine (DAB) (Vector Lab, SK-4105). For viral RNA, we used dsRNA (1:100) (SCICONS, #10010200) and LTL (Vector Lab, #FL-1321) for the proximal tubule marker. Alexa Fluor 555 (Invitrogen, #A31572) and Alexa Fluor 488 (Invitrogen, #21200) were used as the secondary antibody, and nuclei was stained with DAPI (Thermo, #62248). For human kidney sections, immune cells number was recorded as the number of cells per 40X high power field (HPF) then averaged 15 fields containing the maximum number of immune cells. For mice kidney sections, CD4/CD8/F4-80 cells in 40X HPF were recorded then the average for five mice was shown as mean CD4/CD8/F4-80 cells/HPF.

### Statistics and reproducibility
Data represent mean ± SEM in all graphs depicting error bars. Unpaired two-tailed Student's *t*-test was used for comparisons between the two groups. One-way analysis of variance (ANOVA) with Tukey's post hoc test was used to compare multiple groups. The statistical significance of differences between experimental groups was determined using GraphPad Prism 6 and the indicated statistical tests. $P$ values less than 0.05 were considered statistically significant. For mouse experiments, animals were randomly allocated to different groups prior to the experiments. No samples or animals were excluded from the analysis. No statistical method was used to predetermine the sample size. For western blots, each experiment was repeated in at least two independent replicates. For TEs/ERVs analysis, statistical details are provided in the designated method section. All samples were processed in blind fashion. No data were excluded from the analyses.

### Reporting summary
Further information on research design is available in the Nature Portfolio Reporting Summary linked to this article.

## Data availability
Human RNA-seq data are available at Gene Expression Omnibus (GEO) with the accession code GSE115098 and GSE173343. Mouse FA model scRNA-seq data is available at GEO with accession code GSE156686 and can be viewed on the SUSZTAK LAB kidney website (https://susztaklab.com/VisCello/). Mouse UUO scRNA-seq data is available at GEO with accession code GSE182256 and can be viewed on the SUSZTAK LAB website (https://susztaklab.com/UUO/scRNA/). The TE/ ERVs data generated in this study are provided in the Supplementary Information/Source Data file. Further information and requests for resources and reagents should be directed to and will be fulfilled by the lead contact: Katalin Susztak. Email: ksusztak@pennmedicine.upenn.edu Source data are provided with this paper.

## Code availability
Customized code used in the present study is available at github (https://github.com/mulhollandk/ERV_Code).

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

## Acknowledgements

Work in the Susztak laboratory is supported by NIH National Institute of Diabetes and Digestive and Kidney Diseases grants RO1 DK076077 and RO1 DK087635. We thank the University of Pennsylvania Diabetes Research Center (DRC) for the use of the Core services (P30-DK19525). We would like to thank Dr. Avindra Nath from NINDS for sharing the HERV-K plasmid with us.

## Author contributions

This study was led by K.S., P.D., and K.A.M. P.D. performed experiments with assistance from A.V. and J.W. J.P., X.S., A.A., H.L., and H.H. assisted K.M. to perform computational analysis. P.D., K.M., and K.S. wrote the manuscript and all authors approved of the final manuscript.

## Competing interests

The Susztak lab is supported by Boehringer Ingelheim, Lilly, Regeneron, GSK, Merck, Bayer, and Gilead for work that is not related to the current manuscript. The remaining authors declare no competing interests.
