## [Peer Review File · Nature Communications]

Increased levels of endogenous retroviruses trigger fibroinflammation and play a role in kidney disease developmentEditorial Note: The figure on page 7, and the figure on page 15 in this Peer Review File have been amended to remove third-party material where no permission to publish could be obtained.

Reviewers' comments:

Reviewer #1 (Remarks to the Author):

The manuscript "Increased levels of endogenous retroviruses trigger innate immune response and play role in kidney disease and fibrosis" by Dhillon et al details the expression of endogenous retroviruses in diseased human kidneys and in two mouse models of chronic kidney diseases (Unilateral ureteral obstruction and folic acid models). Loss of cytosine methylation associated epigenetic derepression contributes to an increase in endogenous retroviruses levels. In turn, these endogenous retroviruses activate cytosolic nucleotide sensors and contribute to fibroinflammation in the kidney.

Little is known on the expression of endogenous retroviruses, their regulation and potential functional significance; in particular in kidney diseases. The results described in this study fill this gap and give important new insights on the pathogenesis of kidney fibroinflammation. Moreover, these results can contribute to the development of new therapeutic strategies to prevent progression to chronic kidney diseases (CKD).

The manuscript is well-written and the conclusions are well-supported by the data.

Major comments:

- It is unclear for the reader why the tubules were microdissected in human samples but not in mouse tissues? How was the purity of the microdissected tubules controlled? This is important, because epigenetic modifications in proximal tubules and interstitial myofibroblasts are at variance (Hewitson et al PMID: 28611663).
- The data do not provide evidence that additional epigenetic regulators, including regulators of histone methylation and telomerase reverse transcriptase, might contribute to the increase in TE and ERV. Mao et al. (DOI 10.15252) recently showed that telomerase reverse transcriptase can activate a subclass of ERV (independent of its telomerase activity) to form dsRNAs, which are sensed by the RIG-1/MDA5-MAV5 signaling pathway and trigger interferon signaling.
- What might be the mechanisms underlying the regulation of DNMT1 in CKD and mouse models?
- Fig 6E-6H and S6E are not innovative. Results are similar to the publication by Zhou (REF 58).

Minor comments:

- How was the extent of interstitial fibrosis in human kidney samples quantified?
- For the correlation between TE and ERV expression in kidney immune cell fractions: please specify that it has been done in the mouse tissues.
- Introduction: CKD is the tenth leading cause of death (L1) but L4 CKD is the fourth major cause of death. Please clarify.
- Introduction, last paragraph: please clarify what is mean by "immune cell fraction". This is explained later in the results.
- The references are not numbered.

Reviewer #2 (Remarks to the Author):

In this manuscript, Dhillon and colleagues associate increased levels of endogenous retroviruses (ERVs) and transposable elements (TEs) with an innate immune response and the pathogenesis of kidney disease and fibrosis. They present data from control and diseased human kidneys and mouse models that suggest higher expression of TEs and ERVs in diseased kidneys, which correlates with loss of cytosine methylation and is experimentally induced by DNA methyltransferase 1 (DNMT1) inhibition. The authors show further that overexpression of HERV-K in murine cultured kidney tubule cells activates the cytosolic nucleotide sensing pathways and triggers an IFN response in a RIG-I and STING dependent fashion. The authors show lastly that RIG-I is an important mediator of renal fibroinflammation in models of chronic kidney disease (CKD) and that kidney fibroinflammation is

ameliorated with reverse transcriptase inhibitor treatment.

The submitted work tests the hypothesis that ERV expression contributes to fibroinflammation in the kidney, via the accumulation of nucleic acids that activate cytosolic nucleotide sensing pathways. This is an intriguing hypothesis that would establish ERV expression as one major factor in the pathogenesis of human diabetic, hypertensive CKD, and one that can be potentially treated with reverse transcriptase inhibitors. However, the presented data fall short of proving it conclusively.

ERV activation has been associated with an extensive set of inflammatory and infectious diseases, ranging from neuroinflammation, intestinal inflammation, systemic autoimmunity, rheumatic autoimmunity, as well as inflammation associated with aging and cancer. However, the cause-and-effect relationship is unclear. The novelty of the current study is that it adds CKD to the list of inflammatory diseases that potentially show ERV activation (and indeed it provides a comprehensive set of expression data from both humans and mice), but it remains descriptive, with no clear mechanism to support the direction of cause-and-effect.

In human samples, expression of ERVs can go in either direction. For example, it appears that more full-length ERVs are downregulated than upregulated in CKD samples. It is unclear how it can be assumed that these changes lead to overall accumulation of ERV nucleic acids. This would have to be proven experimentally. Is there more ERV overall dsRNA or cDNA bound to the sensor?

Can the upregulation of sensors such as RIG-I and STING really be placed downstream of ERV upregulation? Most of these sensors, including RIG-I and STING are IFN-inducible genes, and their upregulation reflects simply the levels of IFN in the system. Just like ERV upregulation, this may be an epiphenomenon, rather than the cause of inflammation.

The upregulation of cytosolic nucleotide sensing pathways and of IFN by ectopic expression of HERV-K in kidney tubule cells is used here to support a causative effect of ERVs. Unfortunately, the critical details of this experiment are missing. For example, what is present in the "pcDNA3.1-HERV-K full-length" plasmid? Is it a natural copy of HERV-K, and which one? Or is it an artificial consensus sequence? Does it have intact open reading frames and for which products? Does it have an intact reverse transcriptase and can HERV-K cDNA be made and bind to DNA sensors? Many HERV-K proteins have been shown to induce IFN upon overexpression independently, including minor proteins such as Rec (PMID: 25896322). Attributing any effects on nucleic acids alone is not supported by the data presented.

The effect of reverse transcriptase inhibitors in the by UUO-induce model of kidney fibrosis is not as strong as would be expected to be taken as proof of reverse transcription triggering disease. The read out is mostly RNA expression measured by qRT-PCR. The differences are in most cases less than 2-fold. Even though they may be statistically significant, less than 2-fold differences may not be biologically significant. By convention, a 2-fold change threshold is typically applied as the minimum significant change in expression analyses. This is particularly the case for qRT-PCR where signal is amplified by 2-fold in each round of amplification. Differences less than one round of amplification are not typically considered significant.

Reviewer #3 (Remarks to the Author):

The main findings described in the manuscript are as follows:

- Transcription of a number of TEs, including ERVs correlates with renal fibrosis in human kidney samples, while transcription of others do not
- Renal expression of TEs and ERVs correlates with expression of cytosolic nucleotide sensors in human CKD as well as in two murine models of renal fibrosis
- Renal expression of TEs and ERVs correlates with renal mRNA expression profiles, characteristic of some immune cells
- In vitro expression of one particular ERV (HERV-K) in human kidney tubule cells activates the RIG-

STING pathway, resulting in a type 1 IFN response

- In vitro treatment of human kidney tubule cells with an inhibitor of cytosine methylation increases TE/ERV expression and activates a type 1 IFN response
- Tubule specific loss of DNA methyltransferase 1 increased expression of some EV/ERVs, activated the type 1 IFN pathway and aggravated murine renal fibrosis in the folic acid (FA) model
- Knockout of RIG-I protected from activation of the type 1 IFN pathway and development of fibrosis in the FA model
- Inhibition of reverse transcriptase protected from activation of the type 1 IFN pathway and development of fibrosis in the unilateral ureteral obstruction (UUO) model of renal fibrosis

The authors conclude from this, that renal expression of ERVs is increased by loss of DNA methylation (due to unknown mechanisms), which then results in activation of the type 1 IFN pathway via cytosolic nucleotide sensors. This in turn aggravates renal fibrosis.

I have some specific comments and suggestions:

Generally, the topic is novel and clinically very important. Overall, however, the paper is unfortunately rather hard to follow and contains several inconsistencies. References are sometimes in the wrong order and are thus hard to find. The results presented do not fully support the conclusion made by the authors.

Importantly, the authors refer to experiments in STING knockout mice several times, which are, however, not presented in the manuscript.

The authors always talk about TE/ERVs in total. This is rather difficult, since expression of different TE/ERVs seems to have rather different effects. Also demethylation correlates with higher expression of some TEs, while it correlates with lower expression of others and does not correlate at all with the majority of TEs included in the analysis (Figure 4B). Furthermore, while the authors provide good evidence, that loss of RIG-I, as well as interference with reverse transcription ameliorate renal fibrosis, it remains unclear, whether this is in any way related to expression of (certain) ERVs.

The authors should in my opinion rather focus on selected ERVs (which they find to be most strongly correlated with fibrosis) and investigate their individual functions. This could be done by ERV specific knockout or transgenic expression in kidney tissue (as e.g. done by Avindra Nath's group, Ref 23 of the manuscript).

What were the 'WT' controls for the genetically complex PAX8rTA-TRE/CreDNMT1F/F mice?

Why was only the FA model assessed in these mice, as well as in the RIG-I knockouts and not the UUO model?

Why were the effects of reverse transcriptase blockade only evaluated in the UUO model and not in the FA model?

Assessment of BUN levels is lacking in the reverse transcriptase blocker study.

Assessment of renal immune cells by e.g. immunohistochemistry or better FACS analysis is completely lacking and should be performed. Otherwise it is hard to claim, that ERVs trigger innate immunity.

In response to the reviewer's suggestion, we performed several important experiments to support the conclusion. We made changes to the text and figures. Now, the revised manuscript contains 10 main figures, 15 supplemental figures, and 7 supplemental tables.

1. Double-stranded RNA (dsRNA) is an important pathogen-associated molecule pattern (PAMP) produced by viruses. Recent reports suggest the accumulation of self-derived dsRNAs can elicit antiviral response via activation of innate immune response (PMID: 26317465, PMID: 26317466). We performed dsRNA immunoprecipitation experiments to show higher dsRNA content in diseased (UUO) kidneys compared to control (**Figure 4A**). Further, we showed this immunoprecipitated dsRNA activates RNA sensors (RIG-I, MDA5, and STING) followed by the antiviral response in kidney tubule cells (**Figure 4B-4D**). We also show the interaction of dsRNA or ERVs with RIG-I in UUO kidneys by CO-IP and reverse CO-IP experiment to validate the ERVs mediated activation of RIG-STING signaling in kidney tubules cells (**Figure 4E and 4F**). These results further validate ERVs mediated activation of RIG-STING signaling pathway leading to renal inflammation.
2. RIG-I and STING belong to the IFN-stimulated gene (ISG) family, but certain cells regulate their expression through IFN-independent mechanisms (PMID: 21478870, PMID: 28195391, PMID: 25572843). To investigate the role of IFN-mediated regulation of ERVs and cytosolic nucleic acid sensors in kidney tubule cells, we treated PT cells with mouse IFN β and found the levels of ERVs and cytosolic nucleic acid sensors remain unchanged (**Supplemental Figure 6**) suggesting IFN-independent role of RNA sensors in kidney tubules. We also used kidney tissue samples from mice injected with anti-IFNG antibody (IFNG neutralizing antibody) and folic acid. We found only minimal changes in levels of ERVs, and levels of TLR7 and STING were reduced due to lower levels of IFNG in mice suggesting IFN-mediated regulation of nucleic acid sensors may be more prevalent in immune cells (express mostly IFNGR1/2) but not in kidney tubule cells (most abundantly express IFNAR1/2) (**Supplemental Figure 7**).
3. As suggested by reviewer 3, we performed IHC to validate the changes in immune cell fractions such as macrophages, CD4 T cells, and CD8 T cells in healthy and diseased kidneys from mice and human samples (**Supplemental Figure 4**). We found a higher immune cell population in diseased kidneys of mice and human samples compared to control. Reverse transcriptase inhibitors treated mice with UUO showed comparative lower CD4, CD8, and macrophages cells compared to UUO kidneys indicating a strong correlation between ERVs abundance and immune cells fractions.
4. To show the role of other epigenetic regulators in mediating ERVs expression, we included correlation plots showing the association between HERV-K and HERV-H expression with other epigenetic regulators (DNMT1, HDAC1, HDAC2, SETDB1, and TRIM28) and TERT. We found HERV-K and HERV-H expression inversely correlated with DNMT1 and other epigenetic repressors (**Supplemental Figure 8**) suggesting that dysregulated epigenome is primarily responsible for ERVs induction in kidney tubules.
5. As requested, we performed UUO surgery on Pax8rtTA-TRE/Cre-DNMT1 f/f and RIG KO mice and showed Pax8rtTA-TRE/Cre-DNMT1 f/f were more susceptible to kidney injury whereas RIG KO mice showed a protective phenotype in UUO CKD model (**Supplemental Figure 11 and 13**).

6. We also generated RIG KO STING KO (Double KO) mice and showed these mice were protected against folic acid-induced kidney nephropathy (**Figure 8**).
7. We also performed folic acid-induced kidney injury on mice injected with reverse transcriptase inhibitor lamivudine (3TC) or control. We found these mice showed improved kidney function with reduced renal inflammation and injury (**Figure 9**).
8. We also formatted all the references according to journal guidelines.
9. We made multiple additional changes as requested by the reviewers. These changes are highlighted in blue in the revised manuscript version.
10. We attached a point-by-point response including additional data to answer the reviewer's comments in the response letter.

We want to thank all the reviewers for carefully evaluating our work and for their insightful comments and suggestions.

Reviewer #1 (Remarks to the Author):

The manuscript "Increased levels of endogenous retroviruses trigger innate immune response and play role in kidney disease and fibrosis" by Dhillon et al details the expression of endogenous retroviruses in diseased human kidneys and in two mouse models of chronic kidney diseases (Unilateral ureteral obstruction and folic acid models). Loss of cytosine methylation associated epigenetic derepression contributes to an increase in endogenous retroviruses levels. In turn, these endogenous retroviruses activate cytosolic nucleotide sensors and contribute to fibroinflammation in the kidney.

Little is known on the expression of endogenous retroviruses, their regulation and potential functional significance; in particular in kidney diseases. The results described in this study fill this gap and give important new insights on the pathogenesis of kidney fibroinflammation. Moreover, these results can contribute to the development of new therapeutic strategies to prevent progression to chronic kidney diseases (CKD).

The manuscript is well-written, and the conclusions are well-supported by the data.

We thank the reviewer for his or her positive comments and for accurately summarizing our work. The important insights of our work: **(1)** we define TE/ERV expression in healthy and CKD human kidney samples using a large human kidney expression dataset and multiple bioinformatics methods, and **(2)** we show the causal role of ERV activation in CKD development by performing *in vitro* and *in vivo* experiments.

Major comments:

- It is unclear for the reader why the tubules were microdissected in human samples but not in mouse tissues? How was the purity of the microdissected tubules controlled? This is important, because epigenetic modifications in proximal tubules and interstitial myofibroblasts are at variance (Hewitson et al PMID: 28611663).

Thank you. We agree that kidney cell types show differences in their epigenome.

1. To define cell type-specific TE/ERV expressions, we performed *in situ* hybridization and immunostaining studies (**Figure 2B and 2C**).

2. To define the cell-type-specific role of ERV/TE *in vitro*, we used primary tubule (PT) cells. We analyzed the role of ectopic expression of HERV-K in tubule cells (**Figure 3**), we transfected PT cells with dsRNA (**Figure 4A-4D**) and we treated PT cells with the DNMT1 inhibitor (AZA) (**Figure 5D-5G, Supplemental Figure 9**).
3. To define the cell-type specific role of TE/ERVs *in vivo*, we generated tubule-specific DNMT1 KO mice (**Figure 6, Supplemental Figures 10 and 11**).

The reason for the differences in microdissection is practical. The human kidney is larger and can be manually microdissected (glomeruli are visible under a stereo microscope) while the mouse glomeruli are small and cannot be well visualized under the stereo microscope. **DATA 1** shows the efficiency of the microdissection (Qiu et al., 2018, PMID: 30275566).

FIGURE REDACTED

- The data do not provide evidence that additional epigenetic regulators, including regulators of histone methylation and telomerase reverse transcriptase, might contribute to the increase in TE and ERV. Mao et al. (DOI 10.15252) recently showed that telomerase reverse transcriptase can activate a subclass of ERV (independent of its telomerase activity) to form dsRNAs, which are sensed by the RIG-1/MDA5-MAV5 signaling pathway and trigger interferon signaling.

Thank you for your comment. We analyzed several important epigenetic regulators; including DNA and histone-modifying enzymes. The expression of multiple epigenetic modifiers inversely correlates with HERV-K and HERV-H levels indicating that they can regulate ERV expression. Please review their correlation with TE and ERVs levels in **Supplemental Figure 8**. In this study, we mainly focused on dissecting the role of demethylation-induced ERV expression and ERV/TE role in kidney disease using primary tubule cells and tubule-specific DNMT1 KO mice. It will be interesting to investigate the role of other epigenetic modulators.

As TERT plays a crucial role in dividing cells, most somatic cells have undetectable telomerase activity due to transcriptional repression (Mao et al.). We found very low levels of TERT expression in human and mouse kidneys with very little change in diseased conditions.

Still, we can't exclude the role of Telomere reverse transcriptase (TERT) in regulating TE/ERVs expression. We have included a sentence in the discussion.

- What might be the mechanisms underlying the regulation of DNMT1 in CKD and mouse models?

Thank you for raising an important point. DNMT1 regulation is not fully understood. Some studies suggest the role of KLF4, especially in podocytes (PMID: 24812666, PMID: 26108068, PMID: 35024974). Another report showed that DNMT1 is a direct target of the Sp1/NFκB-p65 complex in mouse podocytes in diabetes (Zhang et al., 2017: PMID: 28318634). In general, DNMT gene expression can be induced by Ras-c-Jun signaling pathway, Sp1 and Sp3 zinc finger proteins at transcription level, whereas p53, RB and FOXO3a act as transcriptional regulators and corepressors. Post-translational modifications including acetylation and phosphorylation and availability of methyl donors have been reported to mediate protein stability and activity of the DNMTs especially DNMT1 (PMID: 25949795). However, DNMT1 regulation still needs to be explored in CKD.

- Fig 6E-6H and S6E are not innovative. Results are like the publication by Zhou (REF 58).

Thank you for your comment. **(1)** The publication by Zhou et al studied UO injury, while we studied FA injection-induced disease. **(2)** Here we show how increased ERV expression leads to disease development. The publication by Zhou failed to explain the mechanism of RIG-I activation and did not study cytosine methylation and other nucleotide sensors. The work by Zhou et al did not measure ERV expression in the kidney. **(3)** As requested by the other reviewers in the revised version, we performed new experiments and show results of the FA-induced injury in RIG KO STING KO (Double KO) mice (**Figure 8**) and assessed changes in the expression of cytosolic sensors, ISGs, and kidney histology.

Minor comments:

- How was the extent of interstitial fibrosis in human kidney samples quantified?

Thank you. Our local renal pathologist performed an unbiased review of over 1,000 PAS-stained tissue section by scoring 17 parameters including fibrosis.

- For the correlation between TE and ERV expression in kidney immune cell fractions: please specify that it has been done in the mouse tissues.

Analyses of TE and ERV expression and kidney immune cell fractions were performed in 270 (for TEs) and 485 (for HERV) **human kidney tissue samples** and presented in **Figure 2F** (for TEs) and **Figure S2E** (Full-length HERVs).

- Introduction: CKD is the tenth leading cause of death (L1) but L4 CKD is the fourth major cause of death. Please clarify.

Thank you. CKD is the tenth leading cause of death, but the fourth fastest-growing cause of death. We corrected it in the text.

- Introduction, last paragraph: please clarify what is mean by “immune cell fraction”. This is explained later in the results.

Thank you for your suggestion. We have correlated TE/ERVs levels and 10 different kidney immune cell types (CD8 T cells, CD4 T cells, natural killer cells, proliferating lymphocytes, monocytes, dendritic cells, granulocytes, plasmacytoid dendritic cells, basophils, and macrophages) and found strong correlations between TE/ERV levels and kidney macrophages, CD8 T cells, CD4 T cells, and natural killer cell fraction. We included this information in the text.

- The references are not numbered.

Thank you. We numbered the references according to the journal guidelines.

Reviewer #2 (Remarks to the Author):

In this manuscript, Dhillon and colleagues associate increased levels of endogenous retroviruses (ERVs) and transposable elements (TEs) with an innate immune response and the pathogenesis of kidney disease and fibrosis. They present data from control and diseased human kidneys and mouse models that suggest higher expression of TEs and ERVs in diseased kidneys, which correlates with loss of cytosine methylation and is experimentally induced by DNA methyltransferase 1 (DNMT1) inhibition. The authors show further that overexpression of HERV-K in murine cultured kidney tubule cells activates the cytosolic nucleotide sensing pathways and triggers an IFN response in a RIG-I and STING dependent fashion. The authors show lastly that RIG-I is an important mediator of renal fibroinflammation in models of chronic kidney disease (CKD) and that kidney fibroinflammation is ameliorated with reverse transcriptase inhibitor treatment.

The submitted work tests the hypothesis that ERV expression contributes to fibroinflammation in the kidney, via the accumulation of nucleic acids that activate cytosolic nucleotide sensing pathways. This is an intriguing hypothesis that would establish ERV expression as one major factor in the pathogenesis of human diabetic, hypertensive CKD, and one that can be potentially treated with reverse transcriptase inhibitors. However, the presented data fall short of proving it conclusively.

ERV activation has been associated with an extensive set of inflammatory and infectious diseases, ranging from neuroinflammation, intestinal inflammation, systemic autoimmunity, rheumatic autoimmunity, as well as inflammation associated with aging and cancer. However, the cause-and-effect relationship is unclear. The novelty of the current study is that it adds CKD to the list of inflammatory diseases that potentially show ERV activation (and indeed it provides a comprehensive set of expression data from both humans and mice), but it remains descriptive, with no clear mechanism to support the direction of cause-and-effect.

Indeed, here we present a very large and well-annotated human, and mouse kidney dataset and comprehensively annotate and quantify TE and full-length ERV expression and correlate disease severity and changes in kidney immune cells. ERVs identified in the kidney are different from ERVs identified in

other organs or ERVs associated with other diseases. We think studies performed in the brain have little relevance to the kidney.

In research, we use the 9 Bradford Hill criteria for causality analysis. We establish all 9 criteria in our current study. We analyzed the strength of association and its specificity. We performed extensive causality analysis, by transfecting primary kidney tubule cells with either HERV-K (**Figure 3**) or dsRNA isolated from the diseased kidney (**Figure 4**) and we demonstrate the activation of cytosolic nucleotide sensors. Furthermore, we performed in vivo studies and show the role of cytosolic RNA sensors in kidney disease development using RIG KO and RIG KO STING KO (Double KO) (**Figure 8**). We demonstrated that an antiviral viral drug (reverse transcriptase inhibitor) protects from fibrosis in the UUO and FA mice model of CKD (**Figure 9-10 and Supplemental Figure 14-15**).

In human samples, expression of ERVs can go in either direction. For example, it appears that more full-length ERVs are downregulated than upregulated in CKD samples. It is unclear how it can be assumed that these changes lead to overall accumulation of ERV nucleic acids. This would have to be proven experimentally. Is there more ERV overall dsRNA or cDNA bound to the sensor?

Thank you for pointing this out.

dsRNAs are a hallmark of viral replication intermediates that elicits antiviral response via innate immune pathway. Recently, it's been shown that self-derived dsRNA (bidirectional transcription of TE/ERVs) are important PAMPs recognized by RIG-I and MDA5 RNA sensors.

1. We found **higher number of TEs (n=1925) with increased expression in CKD, compared to those with lower expression (n=786) (Figure 2C)**. It is hard to identify full-length ERVs in short-read sequencing data and the total number of annotated ERVs are also small. The relative expression changes do not reflect absolute abundance.

2. We have quantified absolute TE abundance and show higher amount of ERV/TE in kidney disease **Figure 2B**. It appears that total accumulated TEs or dsRNA play role in the innate immune response via activation of cytosolic sensors.

3. We quantified dsRNA content in healthy and diseased mouse kidneys (protocol for dsRNA IP adapted from Gao et al., 2021 PMID: 33778776). We found higher dsRNA content in kidneys of the UUO model of kidney disease (**Figure 4A**).

4. To check if the accumulated dsRNA are indeed potent ligands for the cytosolic nucleic acid sensors, we transfected primary tubule cells with dsRNA and found activation of cytosolic sensors, increase in expression of ISGs and IFN in WT cells but not in STING KO cells (**Figure 4C and 4D**).

5. To validate that the dsRNA isolated from UUO kidneys binds to RIG-I, we performed immunoprecipitation studies using dsRNA antibody followed by immunoblotting with RIG-I to show the binding of dsRNA to RIG-I (**Figure 4E**) which led to the downstream activation of TBK1/IRF3 signaling cascade.

6. To further validate this interaction, we performed reverse CO-IP using UUO kidney lysate and IP with a RIG-I antibody followed by RNA isolation and qRT-PCR. We found enrichment in ERVs RNA bound to RIG-I (**Figure 4F**).

These results establish the role of the dsRNA mediated activation of RIG-STING signaling by TE/ERV and their role in kidney disease development.

Can the upregulation of sensors such as RIG-I and STING really be placed downstream of ERV upregulation? Most of these sensors, including RIG-I and STING are IFN-inducible genes, and their upregulation reflects simply the levels of IFN in the system. Just like ERV upregulation, this may be an epiphenomenon, rather than the cause of inflammation.

Thank you for making this important point. Our data demonstrate that TE/ERV is an upstream regulator of RIG-I and STING.

1. **Figure 3** in our manuscript shows that HERV-K expression can directly activate RIG-I/STING.
2. In the revised manuscript, we show the direct binding of TE/ERV to their sensor by immunoprecipitation studies (**Figure 4E and 4F**).
3. We isolated dsRNA from SHAM and UO kidneys and transfected primary kidney tubule cells with dsRNA. We observed activation of RIG/STING with an increased level in ISG genes. ISGs and IFNs expression were dependent on STING as it was abrogated in STING KO cells (**Figure 4C and 4D**).
4. We treated primary kidney tubule cells with mouse IFN β to check its efficiency to induce RIG-STING (**Supplemental figure 6**). We found IFN β treated PT cells show an increase in IFN-stimulated genes but failed to induce ERVs or RIG-I expression, suggesting that RIG-I/STING are activated by ERVs in kidney tubule cells.
5. We evaluated levels of ERVs and cytosolic sensors in WT, FA, and FA mice injected with neutralizing IFN γ antibody (**Supplemental Figure 7**). We found minimal changes in ERV and RIG-I expressions.

Overall, we demonstrate that the sensors are downstream of TE/ERV expression. While we did not observe sensor-mediated activation of TE/ERVs we cannot exclude such a possibility (as it is not possible to fully exclude a null hypothesis).

The upregulation of cytosolic nucleotide sensing pathways and of IFN by ectopic expression of HERV-K in kidney tubule cells is used here to support a causative effect of ERVs. Unfortunately, the critical details of this experiment are missing. For example, what is present in the “pcDNA3.1-HERV-K full-length” plasmid? Is it a natural copy of HERV-K, and which one? Or is it an artificial consensus sequence? Does it have intact open reading frames and for which products? Does it have an intact reverse transcriptase and can HERV-K cDNA be made and bind to DNA sensors? Many HERV-K proteins have been shown to induce IFN upon overexpression independently, including minor proteins such as Rec (PMID: 25896322). Attributing any effects on nucleic acids alone is not supported by the data presented.

The HERV-K plasmid is an *in silico*-engineered consensus sequence of full-length 9.4-kb long human-specific HERV-K(HML2) proviruses that contain intact ORFs for all the HERV-K(HML2)-encoded (Gag, Pro, Pol, Env, and the accessory Rec) protein. It has an intact open reading frame and reverse transcriptase (published as Dewannieux et al., 2006 PMID: 17077319). We included this reference in the manuscript.

We show the role of RIG-I and STING in the process as we observed lower expression of ISGs and IFNs in HERV-K transfected RIG KO and STING KO PT cells (**Figure 3**).

We also demonstrate the role of reverse transcriptase in disease development. Treatment of mice with reverse transcriptase inhibitor lamivudine protected from kidney fibroinflammation both in the UO and FA kidney injury models (**Figure 9-10 and Supplemental Figure 14 and 15**).

We did not study IFN expression as the outcome.

The effect of reverse transcriptase inhibitors in the by UO-induce model of kidney fibrosis is not as strong as would be expected to be taken as proof of reverse transcription triggering disease. The read out is mostly RNA expression measured by qRT-PCR. The differences are in most cases less than 2-fold. Even though they may be statistically significant, less than 2-fold differences may not be biologically significant. By convention, a 2-fold change threshold is typically applied as the minimum significant change in expression analyses. This is particularly the case for qRT-PCR where signal is amplified by 2-fold in each round of amplification.

Thank you. To follow your suggestions, we repeated the reverse transcriptase inhibitor experiment in the Folic acid model of CKD (**Figure 9**).

Changes in expression of RIG-I, cGAS, and STING were further validated at protein levels followed by downstream signaling molecules such as phosphorylation of TBK1 and IRF3 by western blots. In addition, most ISGs showed a 4-8-fold increase in our real-time PCR.

We have been keenly reading the fibrosis literature and this is the first time we hear that less than 2-fold relative change is not significant, especially when changes are observed in multiple different outcomes, such as gene expression, kidney function, and kidney structural changes. The FDA approves less than 2-fold (just a 40% change) in kidney function change for drug registration.

Reviewer #3 (Remarks to the Author):

The main findings described in the manuscript are as follows:

- Transcription of a number of TEs, including ERVs correlates with renal fibrosis in human kidney samples, while transcription of others do not
- Renal expression of TEs and ERVs correlates with expression of cytosolic nucleotide sensors in human CKD as well as in two murine models of renal fibrosis
- Renal expression of TEs and ERVs correlates with renal mRNA expression profiles, characteristic of some immune cells
- In vitro expression of one particular ERV (HERV-K) in human kidney tubule cells activates the RIG-STING pathway, resulting in a type 1 IFN response
- In vitro treatment of human kidney tubule cells with an inhibitor of cytosine methylation increases TE/ERV expression and activates a type 1 IFN response
- Tubule specific loss of DNA methyltransferase 1 increased expression of some EV/ERVs, activated the

type 1 IFN pathway and aggravated murine renal fibrosis in the folic acid (FA) model

- Knockout of RIG-I protected from activation of the type 1 IFN pathway and development of fibrosis in the FA model

- Inhibition of reverse transcriptase protected from activation of the type 1 IFN pathway and development of fibrosis in the unilateral ureteral obstruction (UUO) model of renal fibrosis

The authors conclude from this, that renal expression of ERVs is increased by loss of DNA methylation (due to unknown mechanisms), which then results in activation of the type 1 IFN pathway via cytosolic nucleotide sensors. This in turn aggravates renal fibrosis.

I have some specific comments and suggestions:

Generally, the topic is novel and clinically very important. Overall, however, the paper is unfortunately rather hard to follow and contains several inconsistencies. References are sometimes in the wrong order and are thus hard to find. The results presented do not fully support the conclusion made by the authors.

Thank you for these positive comments. We have reorganized the manuscript and worked on eliminating inconsistencies.

Importantly, the authors refer to experiments in STING knockout mice several times, which are, however, not presented in the manuscript.

Sorry for the confusion. The STING knock-out mice showed partial protection from acute and chronic kidney disease as published by (Chung et al Cell Metabolism 2019, PMID: 31474566 and Maekawa Cell Reports 2019, PMID: 31665638). The protection in these articles was attributed to the cytosolic release of mitochondrial DNA. In this manuscript, we only studied primary tubule cells isolated from kidneys of STING knock-out mice.

In our view, the partial protection of RIG-I is expected given the presence of other nucleotide sensors, for example, STING. In response to the reviewer's suggestion, we have generated double knock-out mice with genetic deletion of both RIG-I and STING. These animals show complete protection from kidney disease as shown in **Figure 8**.

The authors always talk about TE/ERVs in total. This is rather difficult, since expression of different TE/ERVs seems to have rather different effects. Also demethylation correlates with higher expression of some TEs, while it correlates with lower expression of others and does not correlate at all with the majority of TEs included in the analysis (Figure 4B). Furthermore, while the authors provide good evidence, that loss of RIG-I, as well as interference with reverse transcription ameliorate renal fibrosis, it remains unclear, whether this is in any way related to expression of (certain) ERVs. The authors should in my opinion rather focus on selected ERVs (which they find to be most strongly correlated with fibrosis) and investigate their individual functions. This could be done by ERV specific knockout or transgenic expression in kidney tissue (as e.g. done by Avindra Nath's group, Ref 23 of the manuscript).

Thank you for your comment and important point. We would like to highlight that this is the first description of TE and ERV expression in human and mouse kidney samples. We show the role of demethylation in increasing TE and ERV expression using DNMT1 inhibitor in primary tubule cells (**Figure 5D-5G, Supplemental Figure 9**) and proximal tubule-specific DNMT1 KO mice (**Figure 6 and Supplemental Figure 10 and 11**). We performed *in vitro* analysis establishing the causal relationship between ERV and RIG-I and STING activation and downstream of RIG-I and STING kidney disease development (**Figure 3 and 4**).

Identification of a single causal ERV is an extremely difficult task, given the expression of a large number of ERV changes and it is beyond the scope of the current manuscript. Here we focused on establishing the pathogenic role of ERV/TE in kidney fibroinflammation. Except for the highlighted paper, almost all other papers have avoided the single causal ERV hypothesis. In addition, we think that an overexpression model could be potentially problematic as we would need to overexpress the human ERV in mouse tissue. This is clearly beyond the scope of the current work which includes 10 figures, 15 supplemental figures, and 7 supplemental tables.

We propose that cytoplasmic dsRNA amount is critical for STING and RIG-I activation. To demonstrate the pathogenicity of TE/ERV expression, we performed additional experiments.

1. We found increased dsRNA content in kidneys of diseased mice compared to healthy control. The dsRNA is a potent inducer of the cytosolic sensors (**Figure 4A**).
2. We show the binding of TE/ERV to the nucleotide sensors and following sensor activation (**Figure 4E and 4F**).

For this reason, our study focused on cytosolic nucleotide sensors. We propose that pharmacological inhibition of STING and RIG-I is now feasible and most likely a better strategy for CKD therapeutics development than ERV inhibitors.

What were the 'WT' controls for the genetically complex PAX8rtTA-TRE/CreDNMT1F/F mice?

Thank you for your comment. DNMT1 F/F mice were used as WT control. We made this change in the text as well.

Why was only the FA model assessed in these mice, as well as in the RIG-I knockouts, and not the UUO model?

Thank you. The FA model develops changes in kidney function biomarkers (serum BUN, creatinine), while the UUO mice only show changes in kidney structure (fibrosis) therefore we believe that the FA is a better CKD model.

Nevertheless, in response to the reviewer's suggestion, we repeated the experiments using the UUO model. We have induced kidney disease by UUO surgery in the PAX8rtTA-TRE/CreDNMT1F/F and RIG KO mice and found RIG KO mice were protected from FA-induced nephropathy while PAX8rtTA-TRE/CreDNMT1F/F mice develop more severe kidney injury. We added this information as now **Supplemental Figure 11** for PAX8rtTA-TRE/CreDNMT1F/F and **Supplemental Figure 13** for RIG KO mice with UUO surgery.

Why were the effects of reverse transcriptase blockade only evaluated in the UUO model and not in the FA model?

Thank you. In general, most papers use only a single mouse fibrosis model. The FA model shows changes not only in fibrosis but also in kidney function parameters, which is an important clinical outcome. We have repeated the reverse transcriptase experiments using the FA model. In response to the criticism, we now show the protective role of the reverse transcriptase inhibitor in the FA model. This information is shown in **Figure 9**.

Assessment of BUN levels is lacking in the reverse transcriptase blocker study.

Thank you. The UUO model does not show changes in BUN as it is a unilateral model of fibrosis, it models obstructive uropathy.

Assessment of renal immune cells by e.g. immunohistochemistry or better FACS analysis is completely lacking and should be performed. Otherwise it is hard to claim, that ERVs trigger innate immunity.

Thank you.

In response to the reviewer's comment, we performed immunostaining studies on human and mouse kidney tissue sections and show marked changes in kidney macrophages, CD4 T cells, and CD8 T cells (**Supplemental Figure 4**).

While tissue FACS analysis could be an important tool, immune cell analysis by FACS could be biased by the digestion method used to liberate the immune cells. Bulk gene expression analysis does not suffer from cell drop-out. Here we performed in silico deconvolution studies using single-cell gene expression and bulk gene expression data. Here we calculated cell fraction changes in 485 Human kidney samples. It is simply not feasible to analyze 13 Immune cell populations in 485 human kidney samples by FACS.

Here, we validated FACS-based immune cell quantification in the mouse UUO model. FACS-based cell quantification strongly correlated with our in silico deconvolution method (Doke et al., 2022, PMID: 3552540).

FIGURE REDACTED

REVIEWER COMMENTS

Reviewer #2 (Remarks to the Author):

In the revised version of their manuscript, Dhillon and colleagues address many of the concerns raised during the initial review. Overall, the manuscript is significantly improved. However, the main concerns regarding direct demonstration of a causative effect of ERVs remain.

The authors demonstrated that ectopic expression of HERV-K in kidney tubule cells causes upregulation of cytosolic nucleotide sensing pathways and of IFN (Figure 3). In response to Reviewers' requests, the authors provided the methodological details of this critical experiment, which were previously omitted. The authors appear to have used the "Phoenix" virus for this experiment. This is an artificial construct, which was designed to reinstate the replicative capacity of the putative ancestral HERV-K "progenitor" by correcting the mutations sustained by genomic HERV-K. It produces infectious particles that re-infect via an extracellular pathway. This infectious virus does not represent defective ERVs and unfortunately cannot be used as such.

In the absence of direct demonstration of ERV causation in kidney disease, the authors should avoid overstating correlative data.

Reviewer #3 (Remarks to the Author):

The authors present a revised version of their manuscript.

While they have addressed most of my minor points, my main concern remains.

The manuscript is unfortunately still rather hard to read and contains several inconsistencies. Most importantly, the data as presented do not fully support the conclusions drawn by the authors.

The authors analyze expression of transposable elements (TE) and endogenous retroviruses (ERV) in human and mouse renal tissues of fibrotic diseases. They find that some of the TEs and ERVs correlate with renal fibrosis, while others do not. Overall, with all due respect for the authors work, one might call this a rather random pattern.

The authors provide in vitro evidence, that some ERVs activate cytosolic nucleotide sensing pathways. It remains unclear though, whether this is a general effect. More importantly, it remains unclear, whether the particular ERVs, which are expressed in renal disease and positively correlate with fibrosis would also do this in vitro and in vivo (while the ERVs which negatively correlate should not).

Next, the authors analyze effects of deficiency of the key intracellular nucleotide sensors RIG-I and STING. However, the effects of RIG-I knockout have already been published in the same models as the authors have used (Zhou J Mol Med 2020, Reference 42). Also renal effects of STING knockout have already been broadly analyzed and published twice. These data are thus only confirmatory in nature.

Furthermore, and most importantly, the authors do not provide any in vivo link between renal ERV expression and the RIG-I and STING pathways. Previously, it has been concluded that mitochondrial DNA plays a role (rather than ERVs) for activation of cytosolic nucleotide sensors. It is thus unclear, whether indeed ERVs activate STING and RIG-I in renal fibrotic diseases or whether they are activated by alternative mechanisms not including ERVs.

As suggested before, I thus believe, that the authors should rethink the general design of their study and specifically address the role of one or two particular ERVs that are over-expressed in murine and human renal disease and also positively correlate with fibrosis.

Finally, in the title of the manuscript, the authors claim, that ERVs trigger 'innate' immune responses. However, they do not provide any compelling evidence for this concept. They show, that in RIG-I and STING knockout mice, renal infiltration of immune cells is increased. This analysis, however, remains superficial and shows that CD4 and CD8 pos. T cells as well as macrophages infiltrate. It is unclear, whether these cells are activated and produce cytokines or whether they are even anti-inflammatory (e.g. IL-10 producers or Tregs) or tissue reparative. The macrophage phenotype remains unclear as well (pro-inflammatory, pro-fibrotic or tissue repair macrophages?) Furthermore, T cells are part of the adaptive immune response and not innate. It is thus unclear why the authors claim in the title of the manuscript, that ERVs activate predominantly innate immune players.

In sum, while the study in general addresses an interesting topic, I do not feel that the data presented by the authors allow to draw the conclusion that ERVs are generally pathogenic and activate innate immune cells via RIG-I and STING pathways to aggravate fibrotic renal diseases.

Reviewer #4 (Remarks to the Author):

Authors have addressed the original Reviewer#1 questions.

REVIEWER COMMENTS

We want to thank all reviewers for their suggestions and comments.

Reviewer #2 (Remarks to the Author):

In the revised version of their manuscript, Dhillon and colleagues address many of the concerns raised during the initial review. Overall, the manuscript is significantly improved. However, the main concerns regarding direct demonstration of a causative effect of ERVs remain.

The authors demonstrated that ectopic expression of HERV-K in kidney tubule cells causes upregulation of cytosolic nucleotide sensing pathways and of IFN (Figure 3). In response to Reviewers' requests, the authors provided the methodological details of this critical experiment, which were previously omitted. The authors appear to have used the "Phoenix" virus for this experiment. This is an artificial construct, which was designed to reinstate the replicative capacity of the putative ancestral HERV-K "progenitor" by correcting the mutations sustained by genomic HERV-K. It produces infectious particles that re-infect via an extracellular pathway. This infectious virus does not represent defective ERVs and unfortunately cannot be used as such.

In the absence of direct demonstration of ERV causation in kidney disease, the authors should avoid overstating correlative data.

Thank you for the comment and we are sorry for the misunderstanding.

1. As requested by the reviewer, to further confirm the direct role of kidney-specific ERVs in activating cytosolic nucleotide sensors, we cloned five naturally occurring ERVs sequences that were upregulated and downregulated in diseased kidneys. We in vitro transcribed HERV RNAs from these clones and transfected PTECs and HKC-8 cells. We found that the RNA sensing pathway (RIG-I/MDA5) was activated in tubule cells upon HERV RNA transfection leading to higher ISGs levels. Interestingly, HERV 1132 which was found downregulated in diseased kidneys had minimal effect on the RNA sensing pathway (**Figure S5B-S5D**). These results strongly support our conclusion that the accumulation of upregulated HERVs activate the cytosolic sensing pathway leading to inflammation during kidney disease.

2. We also would like to add that in the manuscript we did not state that ERV pathogenicity is purely linked to defective ERVs expression, we also show:

- a) ERV protein expression (**Figure 2C and 2D**).
- b) The effectiveness of antiretroviral drug such as lamivudine (**Figures 9 and 10**).
- c) In addition, HERV-K plasmids have been consistently used to demonstrate the pathogenic role of ERVs (PMID: 25926654, PMID: 28330477, PMID: 17257061, PMID: 35420440, PMID: 10516026). Furthermore, the same construct has been used to show the effect of antiviral drugs establishing their viral potency at the nucleotide level (PMID: 28330477).

3. We also would like to highlight that our conclusions are not based on solely using this construct.

- a) We isolated viral dsRNA and show the direct binding of endogenous viral RNA to the nucleotide sensor RIG-I (**Figure 4F**). We also showed dsRNA isolated from diseased kidney activates RIG-I/STING pathway in tubule cells.

Reviewer #3 (Remarks to the Author):

The authors present a revised version of their manuscript.

While they have addressed most of my minor points, my main concern remains.

The manuscript is unfortunately still rather hard to read and contains several inconsistencies. Most importantly, the data as presented do not fully support the conclusions drawn by the authors.

The authors analyze expression of transposable elements (TE) and endogenous retroviruses (ERV) in human and mouse renal tissues of fibrotic diseases. They find that some of the TEs and ERVs correlate with renal fibrosis, while others do not. Overall, with all due respect for the authors work, one might call this a rather random pattern.

We disagree with the reviewer's use of the word "random" in this context. We do present a large body of work here with 10 figures and 16 supplemental figures using multiple transgenic and knock-out lines, in vitro studies, and patient data all of which support our conclusions. We summarize some key points below:

1. Here we used statistics to distinguish random patterns from patterns that are likely different from random. We present a comprehensive omics dataset and rigorously performed linear regression analysis, which has been adjusted to major confounders to show that our observation is not a random pattern.
2. We have discussed Hill's criteria for causality in our rebuttal letter.
3. We found that more TEs (n=1925) with increased expression in CKD, compared to those with lower expression (n=786) (**Figure 2C**).
4. It is hard to identify full-length ERVs in short-read sequencing data and the total number of annotated ERVs is also small. To overcome the fact that the relative expression changes do not reflect absolute abundance, we have quantified absolute TE abundance and showed a higher amount of ERV/TE in kidney disease (**Figure 2B**). It appears that total accumulated TEs or dsRNA play role in strong innate immune response via activation of cytosolic sensors.
5. We found higher dsRNA content bound with RIG-I in the kidneys of the UUO model of kidney disease and these dsRNA activates RIG-I/STING pathway in kidney tubule cells (**Figure 4**).
6. In addition, a strong correlation between upregulated TE/HERVs (that were strongly correlated with fibrosis) with cytosolic nucleotide sensing pathway in both human and mice eliminate the possibility of a random pattern (**Figure 2E** and **Supplemental Figure 2E**).

The authors provide in vitro evidence, that some ERVs activate cytosolic nucleotide sensing pathways. It remains unclear though, whether this is a general effect. More importantly, it remains unclear, whether the particular ERVs, which are expressed in renal disease and positively correlate with fibrosis, would also do this in vitro and in vivo (while the ERVs which negatively correlate should not).

Thank you for the comment. To further confirm the direct role of kidney-specific ERVs in activating cytosolic nucleotide sensors, we cloned five naturally occurring ERVs sequences (that were found upregulated and downregulated in diseased kidneys). We in vitro transcribed HERV RNAs from these clones and transfected PTECs and HKC-8 cells. We found that the RNA sensing pathway (RIG-I/MDA5) was activated in tubule cells upon HERV RNA transfection leading to higher ISGs levels. Interestingly HERV 1132 which was found to be lower in diseased kidneys showed minimal effect on the RNA sensing pathway (**Figure S5B-S5D**). These data strongly support the conclusion that the accumulation of upregulated HERVs activates the cytosolic sensing pathway leading to inflammation during kidney disease.

Additionally,

1. We show that loss of methylation increases TE and ERV expression using DNMT1 inhibitor in primary tubule cells (**Figure 5D-5G, Supplemental Figure 9**) and proximal tubule-specific DNMT1 KO mice (**Figure 6 and Supplemental Figures 10 and 11**).
2. We performed in vitro analysis establishing the causal relationship of ERV with RIG-I/STING activation and downstream of RIG-I/STING kidney disease development (**Figures 3 and 4**).
3. Identification of a single causal ERV is an extremely difficult task, given that the expression of a large number of ERVs change, and it is beyond the scope of the current manuscript. Here we focused on establishing the **pathogenic role of ERV/TE in kidney fibroinflammation**. Except for one paper, almost all other publications have avoided the single causal ERV hypothesis. In addition, we think that an overexpression model could be potentially problematic as we would need to overexpress the human ERV in mouse tissue. Furthermore, transgenic overexpression is usually much higher than endogenously expressed genes.
4. We propose that the cytoplasmic dsRNA amount is critical for RIG-I/STING activation. To demonstrate the pathogenicity of higher viral content, we showed increased dsRNA content in the kidneys of diseased mice compared to healthy control. The dsRNAs are potent inducers of the cytosolic sensors (**Figure 4A**). We show the binding of TE/ERV to the nucleotide sensors following sensor activation (**Figure 4E and 4F**).
5. For this reason, our study is focused on the cytosolic nucleotide sensors. We propose that pharmacological inhibition of STING and RIG-I is now feasible and most likely a better strategy for CKD therapeutics development as compared to ERV inhibitors.

Next, the authors analyze the effects of deficiency of the key intracellular nucleotide sensors RIG-I and STING. However, the effects of RIG-I knockout have already been published in the **same models** as the authors have used (Zhou J Mol Med 2020, Reference 42). Also renal effects of STING knockout have already been broadly analyzed and published twice. These data are thus only confirmatory in nature.

Thank you. Unfortunately, we cannot agree with the reviewer's opinion for the below reasons:

A. Zhou et al studied a different animal model. The publication by Zhou et al studied UUU injury, while we studied FA injection-induced disease in detail.

B. Our manuscript is not about the role of STING and is not related to prior publications that show the role of mitochondrial DNA release in kidney fibrosis.

C. In Zhou et al manuscript, the authors stated "Here, our data showed RIG-I could also be induced in non-viral infection condition. However, the exact mechanism for upregulation of RIG-I other than virus infection needed to be further explored.". This is the exact niche that our work fills.

Furthermore, the following concepts were not mentioned in the Zhou et al paper and thus represent strong advances for our work:

1. ERVs can act as potent inducers for RIG-I/STING activation.
2. We show direct binding of ERV RNA with RIG-I protein in-vivo, by co-IP and reverse co-IP (**Figure 4**).
3. We show a protective phenotype in the FA model and in the UUU CKD mice models (**Figure 7**).
4. We also generated RIG KO STING KO (Double KO mice) and show that these mice are more protected from injury (**Figure 8**).
5. We show the reduced ISG gene expression in RIG KO mice upon injury.
6. We further show that antiviral drug treatment reduces RIG-I/STING activation in diseased kidneys (**Figures 9 and 10**).
7. We show demethylation as a probable cause for TE/ERV induction leading to RIG-I/STING activation using in vivo and in vitro models.

These data altogether shed new light on the mechanism leading to the activation of cytosolic nucleotide sensors in kidneys. These observations are completely novel and were not reported or discussed by Zhou et al. or any other prior publications.

Furthermore, and most importantly, the authors do not provide any in vivo link between renal ERV expression and the RIG-I and STING pathways. Previously, it has been concluded that mitochondrial DNA plays a role (rather than ERVs) for activation of cytosolic nucleotide sensors. It is thus unclear, whether indeed ERVs activate STING and RIG-I in renal fibrotic diseases or whether they are activated by alternative mechanisms not including ERVs.

Thank you and sorry for the misunderstanding. The following in vivo and in vitro experiments and results establish the link between ERVs and RIG-I/STING activation:

1. We show direct activation of the RIG-I/MDA5 pathway upon kidney-specific HERV RNA transfection in tubule cells (**Figure S5B-S5B**) (new).
2. We isolated dsRNA from **diseased** kidneys in mice and showed by co-IP and reverse co-IP that **ERVs RNA were bound to RIG-I** confirming the physical interaction between ERVs and RIG-I in diseased kidneys (**Figure 4**). We also showed colocalization of dsRNA and RIG-I in UOU kidneys (**Supplemental Figure 3F**).
3. We demonstrate that the antiviral drug; Lamivudine reduced viral content in kidneys leading to reduced RIG/STING signaling in FA and UOU mice (**Figures 9 and 10**).

This is the first study that reports the TE/ERVs expression profile in human and mouse kidney diseases. Whilst generating ERV transgenic mice would be interesting, we strongly feel it is outside of the scope of the current manuscript. More importantly, a transgenic mouse line is unlikely to establish the causal role of ERVs in kidney disease as a transgenic line would require human ERVs to be expressed in mouse tissue at a non-physiological level.

Regarding the possible role of mitochondrial DNA, the goal of the project was not to exclude whether other ligands can activate RIG-I and STING. But our data strongly support that TE/ERVs can activate RIG-I and STING in a pathophysiological context.

As suggested before, I thus believe, that the authors should rethink the general design of their study and specifically address the role of one or two particular ERVs that are over-expressed in murine and human renal disease and also positively correlate with fibrosis.

Thank you. As per your suggestions, we selected four top upregulated HERV and one downregulated HERV. These sequences were cloned into vectors and to mimic the in vivo conditions, RNA was transcribed from these naturally occurring HERV sequences. The in vitro transcribed RNA was used to transfect human kidney tubule cells. We found that the upregulated HERV RNA sequences strongly activated the RNA sensing pathways and ISGs unlike the downregulated HERV (**Figure S5B-S5D**). Interestingly, one of HERV (HERV 4321) has more strong antiviral response mediated via the RIG-I sensing pathway and is common in both mice and human tubule cells. The reason for the different responses to different ERVs is intriguing and will form the basis for a further research project (which we must thank the reviewer for) but is clearly outside the scope of the current manuscript.

Finally, in the title of the manuscript, the authors claim, that ERVs trigger 'innate' immune responses. However, they do not provide any compelling evidence for this concept. They show, that in RIG-I and STING knockout mice, renal infiltration of immune cells is increased. This analysis, however, remains superficial and shows that CD4 and CD8 pos. T cells as well as macrophages infiltrate. It is unclear, whether these cells are activated and produce cytokines or whether they are even anti-inflammatory (e.g. IL-10 producers or Tregs) or tissue reparative. The macrophage phenotype remains unclear as well (pro-inflammatory, pro-fibrotic or tissue repair macrophages?) Furthermore, T cells are part of the adaptive immune response and not innate. It is thus unclear why the authors claim in the title of the manuscript, that ERVs activate predominantly innate immune players.

Thank you for your comment, we changed the title and remove “innate”. The word “innate” was included to reference the activation of the **innate immune sensing pathways (RIG-I and STING)**, which we show extensively.

1. As requested by the reviewer, we isolated (sorted) macrophages, NK cells, dendritic cells, and CD8T cells from the kidneys of WT mice after UUO surgery. We have shown the pro- and anti-inflammatory cytokines expression in **Figure S14A-S14D**.

2. Furthermore, we analyzed cytokine and cell activation markers in immune cells by single-cell gene expression in control and UUO mice (**Figure S14E**).

3. Expression of RIG and STING innate immune sensors were found higher in CKD samples of both humans and mice at RNA and protein levels.

4. We show the binding of ERVs with RIG-I in diseased kidneys.

5. Viral RNA (HERV-K/dsRNA) activates RIG/STING pathway-mediated antiviral response which is abrogated in RIG KO/STING KO mice.

6. We also show higher expression of several cytokines such as *Tnfa*, *Il1b*, *Il6*, *Cxcl10*, *Ccl2* and increased level of macrophages markers (*Lyz2* and *Cd68*) in WT FA mice which was found lower in RIG KO mice-injected with FA (**Supplemental Figure 12C**).

In sum, while the study in general addresses an interesting topic, I do not feel that the data presented by the authors allow to draw the conclusion that ERVs are generally pathogenic and activate innate immune cells via RIG-I and STING pathways to aggravate fibrotic renal diseases.

Thank you for seeing the interest in our work and we apologize if our work or responses have been unclear at times. We hope following this response, we have been a bit clearer, and the reviewer can see that the conclusions we draw are strongly supported. Below, we summarize some highlights of our work and concerns that we would like the reviewer to consider if they are still unsatisfied.

The highlights of our work:

This is the first study to show TE and HERV profiles in human and mouse CKD samples. We show demethylation as a probable cause for TE/ERV activation during kidney injury using DNMT1 KO and DNMTi (AZA). We found higher dsRNA in diseased kidneys of mice and dsRNA were bound to RIG-I in diseased kidneys. We also demonstrated that ERVs directly activate the RIG/STING pathways in kidney tubule cells using HERV-K/dsRNA. Further, RIG and STING deficiency protected mice from kidney injury. We present an enormous amount of human kidney data, ERV expression, expression of cytosolic sensors, and immune cells in hundreds of human kidney samples. We analyzed 2 different mouse kidney disease models and multiple genetic knock-outs, including *Dnmt1*, STING, RIG-I, and double knock-outs. With this data, we conclude that ERVs activate RIG-I/STING pathway and contribute to renal fibro-inflammation.

Here are our concerns:

One argument the reviewer repeats is that a single ERV must be responsible for kidney disease. This is not something we state in this manuscript and this is not likely to be the case.

1. We believe that it is the accumulated TE/ERVs content that is responsible for RIG-STING activation leading to an antiviral response. It is highly unlikely that a single ERV's viral potency (to activate RIG-I/STING) would be more than accumulated dsRNA levels found in diseased kidneys.

2. Here, we focused on establishing the pathogenic role of ERV/TE in kidney fibroinflammation. Except for one paper (PMID: 26424568), almost all other papers have avoided the single causal ERV hypothesis as it took more than 10 years of research to determine and select the single causal ERV for that paper. We must keep in mind that showing the pathogenicity of a single ERV will not exclude the pathogenicity of other ERVs.

3. In addition, a mouse overexpression model could be highly problematic as we would need to overexpress the human ERV in mouse tissues. Most overexpression models are artificial as the expressed RNA amount is supra-physiological.

Reviewer #4 (Remarks to the Author):

Authors have addressed the original Reviewer#1 questions.

REVIEWERS' COMMENTS

Reviewer #3 (Remarks to the Author):

In their newly performed experiments, the authors show in vitro, that certain HERVs, which correlate with renal fibrosis in CKD can trigger cytosolic nucleotide sensors in tubulus cells. Whether they also trigger other pathways or whether these HERVs would also do so in vivo remains unclear. Also it remains unclear, whether this activation of nucleotide sensors indeed plays any role in renal fibrosis (and if so, how?- do the transfected tubulus cells produce collagens? do they produce chemokines ? How could these in vivo penetrate the tubular basement membrane? Do the tubulus cells undergo necrosis/apoptosis, produce or activate MMPs?).

My main concerns regarding the total concept of the study therefore remain. Most importantly, the authors show, that a number of ERVs positively correlate with fibrosis, while another large part (2/3 of the full length HERVs) negatively correlates and might thus even be protective. This is not further mentioned or investigated in the manuscript and in my opinion, a general statement about pathogenicity of ERVs in renal fibrosis cannot be made.

Finally and unfortunately, the authors did not follow my suggestion to compare the macrophage or T cell phenotypes in WT versus STING/RIG-I knockout mice. Thus it remains unclear, whether changes in innate (or adaptive) immune-responses contribute to the observed amelioration of renal fibrosis.

Reviewer #5 (Remarks to the Author):

In this revised manuscript the authors have significantly improved their manuscript. This is a very challenging area of investigation and the authors have done a very thorough job. The response to the previous critiques are satisfactory to me.

REVIEWERS' COMMENTS

Reviewer #3 (Remarks to the Author):

In their newly performed experiments, the authors show in vitro, that certain HERVs, which correlate with renal fibrosis in CKD can trigger cytosolic nucleotide sensors in tubulus cells. Whether they also trigger other pathways or whether these HERVs would also do so in vivo remains unclear. Also it remains unclear, whether this activation of nucleotide sensors indeed plays any role in renal fibrosis (and if so, how?- do the transfected tubulus cells produce collagens? do they produce chemokines ? How could these in vivo penetrate the tubular basement membrane? Do the tubulus cells undergo necrosis/apoptosis, produce or activate MMPs?).

Thank you for your note.

We show in the manuscript that the expression of ERVs in kidney tubule cells leads to the activation of the cytosolic nucleotide sensors and cytokine expression (IFN, ISG). We show the key role of STING and RIG-I in vivo in fibrosis development using knock-out mice. Most of the ECM and collagen produced in kidney fibrosis are produced by fibroblasts and not by tubule cells. In fibrosis, tubule cells dedifferentiate and secrete cytokines to attract immune cells and activate fibroblasts. Therefore, we would not expect tubule cells to secrete collagen following ERV expression.

My main concerns regarding the total concept of the study therefore remain. Most importantly, the authors show, that a number of ERVs positively correlate with fibrosis, while another large part (2/3 of the full length HERVs) negatively correlates and might thus even be protective. This is not further mentioned or investigated in the manuscript and in my opinion, a general statement about pathogenicity of ERVs in renal fibrosis cannot be made.

Thank you for your comment and sorry for the misunderstanding. Figure 1c shows that level 1,925 ERV is higher in fibrosis and level 786 is lower in fibrosis. In **Figure S5b-S5d**, we show the accumulation of HERVs, the direct binding of ERVs to the nucleotide sensors, the activation of the sensors and the increase in levels of cytokines.

Finally and unfortunately, the authors did not follow my suggestion to compare the macrophage or T cell phenotypes in WT versus STING/RIG-I knockout mice. Thus it remains unclear, whether changes in innate (or adaptive) immune-responses contribute to the observed amelioration of renal fibrosis.

Thank you. We included macrophages and T cell profiles from STING/RIG-I KO SHAM and UUO kidneys in **Supplemental Figure 14a-14d**.

Reviewer #5 (Remarks to the Author):

In this revised manuscript the authors have significantly improved their manuscript. This is a very challenging area of investigation and the authors have done a very thorough job. The response to the previous critiques are satisfactory to me.

We want to thank the reviewers for their encouraging comments.